# Biscuits Enriched with Monofloral Bee Pollens: Nutritional Properties, Techno-Functional Parameters, Sensory Profile, and Consumer Preference

**DOI:** 10.3390/foods12010018

**Published:** 2022-12-21

**Authors:** Rita Végh, Mariann Csóka, Éva Stefanovits-Bányai, Réka Juhász, László Sipos

**Affiliations:** 1Department of Nutrition, Institute of Food Science and Technology, Hungarian University of Agriculture and Life Sciences, 1118 Budapest, Hungary; 2Department of Food and Analytical Chemistry, Institute of Food Science and Technology, Hungarian University of Agriculture and Life Sciences, 1118 Budapest, Hungary; 3Department of Dietetics and Nutrition Sciences, Faculty of Health Sciences, Semmelweis University, 1088 Budapest, Hungary; 4Department of Postharvest, Commercial and Sensory Science, Institute of Food Science and Technology, Hungarian University of Agriculture and Life Sciences, 1118 Budapest, Hungary; 5Institute of Economics, Centre of Economic and Regional Studies, Lóránd Eötvös Research Network, 1097 Budapest, Hungary

**Keywords:** in vitro antioxidant capacity, texture analysis, CIELAB colour space, quantitative descriptive analysis, consumer acceptance, penalty analysis, preference map

## Abstract

Bee pollens are potential functional food ingredients as they contain essential nutrients and a wide range of bioactive compounds. The aim of this study was to investigate the effects of enrichment with monofloral bee pollens on the nutritional properties, techno-functional parameters, sensory profile, and consumer preference of biscuits. Biscuits were prepared according to the AACC-approved method by substituting wheat flour with pollens of rapeseed (*Brassica napus* L.), phacelia (*Phacelia tanacetifolia* Benth.) and sunflower (*Helianthus annuus* L.) at 2%, 5% and 10% levels. The macronutrient composition of the biscuits was determined: crude protein content (Kjeldahl method), crude fat content (Soxhlet extraction), ash content (carbonization), moisture content (drying), carbohydrate content (formula). Their total phenolic content (TPC) and in vitro antioxidant capacity (FRAP, TEAC, DPPH) were determined spectrophotometrically. The colour of the biscuits was measured using a tristimulus-based instrument, and their texture was characterized by using a texture analyser. Sensory profile of biscuits was determined by qualitative descriptive analysis (QDA). The consumer acceptance and purchase intention of the biscuits were also evaluated, based on the responses of 100 consumers. Additionally, an external preference map was created to illustrate the relationship between consumer preference and the sensory profile of the biscuits, and penalty analysis was conducted to identify directions for product development. Phacelia pollen appeared to be the most effective for improving the nutritional quality of biscuits. The addition of phacelia pollen at the 10% substitution level increased the protein content and TPC of the control biscuit by 21% and 145%, respectively. Significant changes (*p* < 0.05) were also observed regarding the colour and texture of biscuits. The results of the QDA revealed that biscuits containing pollens of different botanical sources have heterogeneous sensory attributes. The biscuit containing sunflower pollen at the 2% substitution level was preferred the most (overall liking = 6.9 ± 1.6), and purchase intentions were also the highest for this product. Based on the results of the present study, it is recommended to use sunflower pollen for developing pollen-enriched foods in the future.

## 1. Introduction

There is a growing trend towards consuming foods enriched with health-beneficial substances of natural origin. Apicultural products, including bee pollens, are well applicable for this purpose, as they contain various macro- and micronutrients that are essential for or have beneficial effects on the human body [1]. Bee pollen is created by honeybees (*Apis mellifera* L.), which moisten pollen grains with nectar and salivary secretions, then transfer the formed pellets to the hive. Pollen pellets can be harvested by using pollen traps placed at the entrance of the hives [2]. Approximately 1500 tonnes of bee pollen are produced every year worldwide [3], and this is expected to grow extensively in the coming years [4]. Bee pollen plays a relatively significant economic role in Argentina, Brazil, China, Spain and Hungary [5].

The nutritional and sensory properties of pollens depend on their botanical composition [6,7,8]. Pollen is a good source of lysine [2,9], which is the first limiting amino acid in the major cereal species, including wheat [10]. The concentration of α-linolenic acid is typically higher compared to linoleic acid in pollen, suggesting that it might help to regulate the balance of n-6/n-3 polyunsaturated fatty acids in human diets [11]. Macro- and microelements, vitamin B complex, vitamin C, and vitamin E are also present in bee pollens in considerable amounts [12,13]. Additionally, pollen is a rich source of antioxidants, mainly including phenolic acids, flavonoids, and carotenoids [2,5,14]. Given the nutritional properties of bee pollen, it is appropriate for natural dietary supplementation [2,3]. It can also be used as a therapeutic product owing to its antioxidant, anti-inflammatory, anticarcinogenic, antibacterial, hepatoprotective and anti-atherosclerotic potential [15]. Moreover, owing to their favourable physicochemical composition and techno-functional properties, bee pollens can be applied effectively as a functional food ingredient [16].

Biscuits are popular bakery products, primarily because they can be consumed quickly, and they have a long shelf life and a varied taste and texture. The essential ingredients of these products are flour, fat/oil, sugar, water, and chemical leavening agents, such as baking soda. These ingredients are considered to be unhealthy by consumers; therefore, a large number of studies have been conducted in recent decades in order to improve the nutritional quality of biscuits [17]. Bee pollens have been widely used by researchers as a functional ingredient of foods including bakery products [6,18,19,20,21]. The source plants of bee pollens used for enrichment were identified in only a few studies, although, the physical, chemical, and sensory properties of pollens are strongly influenced by their botanical origin [6]. The objective of this work is to compare biscuits enriched with monofloral bee pollens of rapeseed *(Brassica napus* L.), phacelia (*Phacelia tanacetifolia* Benth.) and sunflower (*Helianthus annuus* L.), based on their nutritional properties, colour, texture, sensory profile, consumer preference and purchase intention. Substitution levels of 2, 5, and 10% were chosen in order to obtain results that are comparable with literature data. Pollens of the selected plant species can potentially be produced monoflorally because they are grown as monoculture, are very attractive pollen sources for honeybees, and have economic significance in several countries.

## 2. Materials and Methods

### 2.1. Raw Materials of Biscuits

Commercial wheat flour (BL 55, Gyermelyi Zrt, Gyermely, Hungary), ground sucrose (Magyar Cukor Zrt, Kaposvár, Hungary), margarine with a fat content of 70% (Bunge Polska Sp, Kruszwica, Poland), glucose (Dénes-Natura Kft, Pécs, Hungary), salt (Salzwelten GmbH, Hallstatt, Austria), and baking soda (Házi Piros Paprika Kft, Sükösd, Hungary) were purchased from a retail store in Hungary. Dried honeybee-collected bee pollens originating from rapeseed, phacelia and sunflower were provided by a local beekeeper. These products were harvested between April and July 2021 in Pest county, Hungary. Pollen loads were sorted by colour, shape, and size to remove pellets of unknown botanical origins. Subsequently, the botanical composition of these products was identified by microscopic pollen analysis. Margarine and glucose syrup were stored in 4 ± 0.5 °C, while other ingredients were stored in a dark place at room temperature until use.

### 2.2. Pollen Identification

Microscopic pollen analysis was used to identify the botanical origin of pollen pellets. The determination was performed by an expert of the Melissopalynological Group of the International Honey Commission. After homogenization, ten pollen loads were selected randomly and suspended and dispersed in 10 mL distilled water. Subsequently, 30 µL of the suspension was transferred onto two slides using a micropipette. After drying on a hot plate, they were covered with glycerin gelatin mixture or glycerin gelatin mixture stained with fuchsine. Pollen grain identification was performed for both slides by examining the entire area of a 20 mm × 20 mm cover slip [22]. Pollen grains were identified based on their specific morphological characteristics, by using a database which included the microscopic images pollen grains originating from more hundred plant species that are present in Hungary. DELTA Optical binocular light microscope (Delta Optical, Warsaw, Poland) at a 400× magnification was used for the determination.

### 2.3. Biscuit Preparation

Biscuits were prepared according to the AACC-approved method 10–50D [23] using the recipe presented in Table 1. The control sample contained ingredients included in the standard. Ground bee pollens were used to substitute 2, 5 or 10% of the wheat flour. All ingredients were weighed in a plastic bowl with a precision of two decimal places, then mixed into a homogenous mass. The dough was then sheeted to a thickness of 7 mm. Biscuits with a diameter of 50 mm were formed and baked in an electrically heated rotary oven (Gierre, Milano, Italy) for 10 min at 205 °C. Samples were then cooled for 30 min at room temperature, and packed in sealable plastic bags. Samples used for chemical analysis were stored at −20 ± 2 °C, while samples used for the determination of texture, colour or sensory attributes were stored at room temperature for a maximum of 24 h. Biscuits prepared accordingly can be seen on Figure 1.

### 2.4. Determination of Macronutrients

For crude protein determination, the classical Kjeldahl method was applied. A nitrogen-to-protein conversion factor of 6.25 was used during the calculations. Crude fat content was determined by Soxhlet extraction using petroleum ether as a solvent. Ash content was determined gravimetrically by carbonization at 525 ± 25 °C in a laboratory furnace until constant weight is achieved. Moisture content was also determined gravimetrically by drying biscuits at 105 ± 2 °C in a laboratory air-oven until constant weight. The total carbohydrate content of the pollens was calculated using the following formula (Equation (1)):(1)Carbohydrate (%)=100−(Moisture (%)+Crude protein (%)+Crude fat (%)+Ash (%)) 

### 2.5. Determination of the Total Phenolic Content and In Vitro Antioxidant Capacity

#### 2.5.1. Extract Preparation

Extracts of biscuit samples were prepared as follows: 1.00 g of ground and homogenized biscuits were weighed in centrifuge tubes and were dissolved in 10 mL of solvent. Distilled water was used as a solvent to extract compounds that are water-soluble. The mixtures were homogenized by vigorous shaking for 30 s, then treated in an ultrasonic bath (TESLA TYP: UC003 B81, 300 W, 40 °C) for one hour. Subsequently, samples were centrifuged (Hettich Holding GmbH, Kirchlengern, Germany) at 11,000 rpm for 10 min. Then, 1.5 mL of supernatants were transferred into Eppendorf tubes and stored at −20 ± 2 °C until analysis.

#### 2.5.2. Total Phenolic Content

For the determination of the total phenolic content (TPC), the Folin–Ciocalteu method developed by Singleton and Rossi (1965) was used [24]. Firstly, 1250 μL of distilled water:Folin–Ciocalteu reagent (90:10) solution was pipetted into test tubes, and 150 μL of methanol:distilled water (80:20) was added. Subsequently, 100 μL of sample extract was added to the mixture. After one minute, 1000 μL of Na_2_CO_3_ solution (0.7 M) was also added. Test tubes were vortexed, and warmed in a 50 °C water bath for 5 min. Absorbances of the solutions were measured at 760 nm against a blank solution. The results are expressed in mg GAE (gallic acid equivalent)/100 g dry weight.

#### 2.5.3. Ferric Reducing Antioxidant Power (FRAP) Assay

The FRAP assay was conducted as proposed by Benzie and Strain (1996) [25]. The FRAP reagent was prepared by mixing sodium acetate buffer (300 mM/L, pH = 3.6), iron(III)chloride (20 mM/L) and 2,4,6-tri-2-pyridinyl-1,3,5-triazine (TPTZ) (10 mM/L) in a 10:1:1 ratio. For the determination, 1500 μL of FRAP reagent was pipetted in test tubes, then 50 μL of sample extract was added. After 5 min, absorbances of the solutions were measured at 593 nm against a blank solution. The results are expressed in mg AAS (ascorbic acid equivalent)/100 g dry weight.

#### 2.5.4. Trolox Equivalent Antioxidant Capacity (TEAC) Assay

The TEAC assay was performed by applying the method of Miller and co-workers (1993) [26]. As a first step, the peroxyl radical was prepared by mixing 39.2 µL of potassium persulfate (125 mM) and 1960.8 µL of ABTS solution (7 mM). The radical was stored in the dark at room temperature for one day. Subsequently, it was diluted 80-fold with phosphate buffer (pH = 7.4), and its absorbance was adjusted to 0.700 ± 0.002 at 734 nm. Then, 1950 µL of ABTS was pipetted into test tubes, and 40 µL of the sample extract was added. After shaking for five minutes, absorbances were measured at 734 nm against the phosphate buffer. The results were expressed in mg TE (trolox equivalent)/100 g dry weight.

#### 2.5.5. 2,2-Diphenyl-1-picryhydrazyl (DPPH) Assay

The measurement of DPPH radical-scavenging activity was carried out according to the method of Blois (1958) [27] and modifications by Hatano and co-workers (1988) [28]. Reagent was prepared by dissolving 9 mg of 2,2-diphenyl-1-picryhydrazyl (DPPH) in 100 mL of methanol in a dark glass bottle. Solutions were prepared by mixing 1000 µL of DPPH reagent, 800 µL of distilled water and 200 µL of sample extract into sealable test tubes. Solutions were stored in the dark for 30 min, and their absorbances were measured at 517 nm against distilled water. The results were expressed in mg TE (trolox equivalent)/100 g dry weight.

### 2.6. Spectral Colour Measurement

The colour parameters of the baked biscuits were measured using a Konica Minolta chroma meter CR-410 device (Konica Minolta, Inc., Tokyo, Japan). Results are expressed as CIELAB colour coordinates, where L* indicates the lightness from black (0) to white (100), while a* describes the red–green colour (a* > 0 indicates redness, a* < 0 indicates greenness), and b* describes the yellow–blue colour (b* > 0 indicates yellowness, b* < 0 indicates blueness) [29]. Hue is a qualitative colour parameter that refers to an angular position around a point or axis on a colour space coordinate diagram. Chroma (saturation) is a quantitative colour parameter that can be defined as the strength of a hue. Hue angles (h°) and chroma (C*) values of the samples were calculated by using the following formulas (Equations (2) and (3)):(2)$$h{}^{\circ}=tan^{-1}\;\frac{*}{a*}$$
(3)$$C^{*}=\sqrt{\left(a{*}^{2}+b{*}^{2}\right)}$$

### 2.7. Baking Loss, Geometry

After cooling to room temperature, the dimensions of biscuits were measured. Their volumes were calculated by the following equation (Equation (4)), where r is the radius and h is the height of biscuits:(4)$$V=\pi \times r^{2}\times h$$

The weight of the biscuits was measured with an accuracy of four decimal places before and after baking. Baking loss (%) was determined using the following equation (Equation (5)), where w_1_ is the weight of the biscuit prior baking (g) and w_2_ is the weight of the biscuit after baking (g):(5)$$Baking\;loss\;\left(\%\right)=\frac{w_{1}-w_{2}}{w_{1}}\times 100\;$$

### 2.8. Texture Analysis

The texture of the biscuits was characterized by applying a Brookfield CT3 Texture Analyzer (LFRA 4500 Texture Analyzer, Brookfield, WI, USA). Nine biscuits were selected randomly from each type. Measurements were conducted on the centre of the selected biscuits. The texture profile analysis was performed with a TA44 probe (stainless steel cylinder; diameter: 4 mm). The following test parameters were adjusted: total cycles: 2; test speed: 1 mm/s; target value: 4 mm; trigger load: 4 g. Data recording and analysis of the texture profile were performed using TexturePro CT v1.9 build 35 software (Ametek Brookfield, Middleborough, MA, USA). The units for certain texture parameters are given in the default form provided by the software. Based on the texture profile, the following parameters were determined: hardness (g), adhesive force (g), fracturability (g), quantity of fractures (-), cohesiveness (-) and springiness (mm). Gumminess (g) and chewiness (mJ) were calculated using the following equations (Equations (6) and (7)):(6)Guminess (g)= hardness (g)×cohesiveness
(7)Chewiness (mJ)=guminess (g)×springiness

### 2.9. Sensory Tests

Sensory tests were carried out in a sensory laboratory (Hungarian University of Agriculture and Life Sciences, Institute of Food Science and Technology, Department of Postharvest, Commercial and Sensory Science) that met the standard requirements [30]. The work was conducted in accordance with The Code of Ethics of the World Medical Association (Declaration of Helsinki). The tests were carried out anonymously and on a voluntary basis. Participants gave informed consent via the statement “I am aware that my responses are confidential, and I agree to participate in this experiment”, where an affirmative reply was required to enter the test. They were able to withdraw from the experiment at any time without giving a reason. The products tested were safe for consumption. Before the test, participants were informed that the biscuits contained pollen and gluten, which may cause hypersensitivity reactions in sensitive individuals.

#### 2.9.1. Sensory Profile Analysis

Sensory profiling was conducted by 12 trained panellists (6 females and 6 males, between the ages of 20 and 28) with the necessary knowledge and experience in sensory descriptive analysis including techniques and practices in attribute identification and terminology development. The members of the trained sensory panel were practiced and highly skilled in sensory profiling of bakery products. Each panellist went through training that met the standard requirements [31,32]. The performance of the trained sensory panel was analysed using the mixed assessor model–control of assessor performance (MAM-CAP) table method for discrimination, agreement, repeatability, and scaling. Sensory tests were performed using the quantitative descriptive analysis (QDA) method [33]. The panel evaluated the biscuits using a scale between 0 and 100 for each, where 0 was the lowest and 100 was the highest score. Panellists analysed 32 attributes of the biscuits regarding appearance, odour, texture, and taste. A separate text box was available to describe other attributes. To prevent sensory fatigue, there was a two-hour break between appearance/odour attributes and taste/texture attributes. Tests were conducted using two replicates to ensure data reliability. Products were coded with random, three-digit numbers starting with non-zero. As a taste neutralizer, mineral water was provided for each panellist, which had constant composition and neutral taste.

#### 2.9.2. Consumers’ Preference Tests

The tests were designed and implemented according to the international standard for consumer preference tests [34,35]. A total of 100 consumers participated from the young age group (between 18 and 30 years), comprising 65 females and 35 males. Inclusion criteria were as follows: willingness to participate in the experiment, regular biscuit eater, does not have a history of food allergies or intolerances including sensitivity to beekeeping products. Each participant evaluated biscuits on two sessions to minimize fatigue. Products were coded with random, three-digit numbers starting with non-zero. As a taste neutralizer, mineral water was provided for each participant.

Participants recorded their answers on a questionnaire consisting of four parts. Part A recorded information regarding the socio-demographics (age, gender, municipality), total net household income and frequency of biscuit consumption. Part B included a nine-category monotonic ascending hedonic response scale with descriptive terms (1 = dislike extremely, 2 = dislike very much, 3 = dislike moderately, 4 = dislike slightly, 5 = neither like, nor dislike, 6 = like slightly, 7 = like moderately, 8 = like very much, 9 = like extremely) and emoticons for each category. Ten attributes were evaluated: darkness, overall odour intensity, sweet odour intensity, margarine odour intensity, overall taste intensity, sweet taste intensity, margarine taste intensity, hardness, crumbliness and overall acceptance. In Part C, the above-listed properties (except for overall acceptance) were evaluated in a 5-category just-about-right (JAR) scale (−2 = not enough at all, −1 = not enough, 0 = just about right, 1 = too much, 2 = far too much). In part D, a question was included regarding the purchase intention on a 5-point structured scale (1 = definitely would not buy, 2 = probably would not buy, 3 = maybe buy or maybe not buy, 4 = probably would buy, 5 = definitely would buy) [36].

### 2.10. Statistical Analysis

The colour, texture and nutritional parameters were determined in twelve, nine or four parallel measurements, respectively. Data are expressed as mean ± standard deviation (SD). In the case of colour parameters, ANOVA was applied with Tukey-HSD post hoc test to determine statistical differences between biscuits (α = 0.05). In the cases of textural and nutritional parameters, statistical differences were determined by Kruskal–Wallis nonparametric test and Dunn’s pairwise procedure with Bonferroni correction (α = 0.05).

Sensory profile data and consumer preference scores are expressed as mean ± standard deviation (SD). Statistical differences between biscuits were determined for each attribute by applying ANOVA with Tukey-HSD post hoc test (α = 0.05). The performance of the trained sensory panel was analysed using the mixed assessor model–control of assessor performance (MAM-CAP) table method. The MAM-CAP table was created using R-project software version 4.0.2 (The R Foundation, Vienna, Austria) with the MAM-CAP-package [37,38]. Significant differences between samples regarding purchase intentions were determined by the parametric k proportion test (Chi-square test followed by Marascuilo procedure).

The descriptive sensory attributes were submitted to Principal Component Analysis (PCA) and the obtained sensory map was used as a plot for positioning consumer overall liking data grouped into three clusters (PREFMAP) [39]. To identify consumer segmentation, Hierarchical Cluster Analysis (HCA) was run on the overall liking hedonic scores using Euclidean distances and Ward’s method as the agglomeration criterion.

Penalty analysis was run on just-about-right (JAR) data. Firstly, the JAR values were amalgamated into 3 groups. Categories 1–2 were labelled as “not enough”, category 3 as “JAR” and categories 4–5 as “too much”. Then, the mean overall acceptance level of samples was calculated for each group. The penalties (or mean drops) were calculated as the differences between the mean of one of the non-JAR categories and the mean of the JAR category. These values were plotted against the respondent percentage of the non-JAR category, placing each response in a so-called mean drop plot. Each statistical test was carried out at a significance level of 5% by using XLSTAT Sensory Version (Addinsoft, Long Island, NY, USA, 2016).

## 3. Results and Discussion

### 3.1. Botanical Origin of Pollen Samples

The results of the pollen analysis confirmed that the pollens used for enrichment originated from rapeseed (*Brassica napus* L.), phacelia (*Phacelia tanacetifolia* Benth.) and sunflower (*Helinathus annuus* L.). The proportion of the predominant pollen was above 96% in each sample; thus, they can be referred to as monofloral based on the criteria proposed by Campos and co-workers (2008) [9].

### 3.2. Nutritional Properties of Biscuits

#### 3.2.1. Macronutrient Composition

The macronutrient compositions of biscuits are presented in Table 2. The control biscuit consisted of 77.66% carbohydrates, 11.35% fat, 5.43% protein and 1.05% ash. Its moisture content was 4.52%. The addition of rapeseed or phacelia pollens at 10% concentration level increased the crude protein content of biscuits significantly. The reason behind this observation is that the protein content of rapeseed or phacelia pollen is usually above 25%, while sunflower pollen can be characterized by a protein content of approximately 15% [16,40,41]. The substitution of wheat flour with 10% of rapeseed pollen resulted in a statistically significant increasement of ash content. However, no significant differences were observed between the control and pollen-containing biscuits in terms of the carbohydrate and crude fat content. These results are in accordance with the observations of Krystyjan and co-workers (2015) [20].

#### 3.2.2. Total Phenolic Content and Antioxidant Properties

The results of antioxidant tests are presented in Table 3. Examining the trends of the results, pollen substitution tended to increase both TPC and antioxidant capacity (FRAP, TEAC, DPPH) of biscuits enriched with pollen of each plant species (rapeseed, phacelia, sunflower). The statistical results showed a significant increase in TPC for biscuits containing pollens at the 10% substitution level compared to the control, and 5% was also sufficient in the case of phacelia. Similar results were obtained for antioxidant capacity (FRAP, TEAC, DPPH), where 10% supplementation also resulted in a significant increase. However, in the case of DPPH assay, substitution with 5% phacelia pollen was also sufficient for a statistically significant increase. The results described above are in agreement with previous studies [19,20,21]. However, in this experiment, lower absolute values were observed compared to the results of other studies, probably due to the differences between extraction methods. Lawag and co-workers (2020) concluded that mixtures of water and organic solvents are more efficient for the extraction of phenolic compounds from bee pollen than distilled water [42]. In this experiment, distilled water was used as a solvent to obtain results which are better indicators of the processes taking place in the human body.

The increase in these parameters can be attributed to the fact that the total phenolic content of pollens is 30.59 mg GAE/g on average and range between 0.69 and 213.20 mg GAE/g [2]. The results of this study indicate that phacelia pollen increases the TPC and antioxidant capacity of the products to the greatest extent, followed by rapeseed and sunflower. However, based on the results of Dundar and co-workers (2021), the bioaccessibility of phenolic compounds of pollen containing-biscuits is limited [19]. This can be explained by the fact that the cell wall of pollen grains is very complex and resistant, which does not allow intracellular compounds to release [43]. Since the cell walls of pollen grains of different plant species are different [44], the bioaccessibility of pollen nutrients may also be influenced by the botanical origin. Additionally, the bioaccessibility, bioavailability and bioactivity of phenolic compounds are also affected by proteins, lipids and carbohydrates present in food matrices. These molecules may interact with the phenolic compounds, and act as carriers of polyphenols through the digestive tract. Moreover, they can protect polyphenols from oxidation during their passage through the gastrointestinal tract. However, interactions may reduce the nutritional value of phenolic compounds [45]. For example, Kostić and co-workers (2021) observed a significant decrease in total phenolic and flavonoid contents of thermally treated skimmed goat milk enriched with sunflower pollen probably due to the interactions between caprine milk casein micelles and pollen polyphenols [46]. In the view of the above, it is important to conduct further research on the interactions between phenolic compounds and macronutrients of foods enriched with pollens of different botanical sources.

### 3.3. CIELAB Colour Parameters

The measured CIELAB colour coordinates (L*, a*, b*) and calculated parameters (C*, h°) are indicated in Table 4. As pollen is darker than wheat flour, the lightness (L*) of each biscuit was significantly lower compared to the control. It can be observed that the lightness of the biscuits decreased with increasing pollen concentrations. Additionally, the formation of Maillard reaction products is expected to increase with increasing pollen concentration due to the presence of amino acids in pollens. The Maillard reaction is a nonenzymatic browning process, during which brown polymers called melanoidins are formed [47]. The addition of pollen of each plant species resulted in a significant increasement of redness (a*) in biscuits. The yellowness (b*) value also increased with increasing concentrations of rapeseed or sunflower pollens; however, the addition of phacelia pollen resulted in a significant decrease in yellowness in the samples. Flavonoids and carotenoids are major pigments of pollens of most plant species [48]. Presumably, the content and ratio of these pigments are responsible for the increases of redness and yellowness in pollen-containing biscuits. Nevertheless, the colour of phacelia pollen is bluish [48], which led to the decrease in yellowness in biscuits. Chroma (C*) is a parameter indicating the saturation of a colour. Results of this study suggest that the substitution of flour with rapeseed or sunflower pollens increase the chroma value of biscuits enriched with pollens of rapeseed/phacelia, while this parameter is significantly lower in phacelia pollen-containing samples compared to the control. It can be observed that these differences increase with increasing pollen concentration. Hue angle (h°) is a qualitative parameter of colour, which was also affected by pollen addition because the colour of each pollen differed notably from wheat flour.

From the CIELAB colour coordinates, total colour differences (ΔE_ab*_) were calculated for each pair of samples, and are presented in Table 5. According to the literature data, colour differences perceived by the human eye can be grouped as follows: ΔE_ab*_ < 1 “observer does not perceive the difference”, 1 < ΔE_ab*_ < 2 “only an experienced observer can perceive the difference”, 2 < ΔE_ab*_ < 3.5 “unexperienced observer also perceive the difference”, 3.5 < ΔE_ab*_ < 5 “clear difference in colour is perceived”, 5 < ΔE_ab*_ “observer perceives two different colours” [49]. The results of the present study indicate that each biscuit is distinguishable from all others; however, only an experienced observer can perceive the difference between the 5% sunflower pollen-containing and 10% rapeseed pollen-containing biscuits (ΔE_ab*_ = 1.7). Additionally, the colours of biscuits containing 2% rapeseed or sunflower pollen are also hard to distinguish by the human eye (ΔE_ab*_ = 2.1). The 10% phacelia pollen-containing sample differed most from the other samples. There is an overall tendency that the more we increase the amount of pollen, the more the samples deviate from the control. The total colour difference resulting from pollen addition is greatest for phacelia, followed by sunflower and rapeseed pollens.

### 3.4. Geometric Properties, Baking Loss

The results of geometric attributes are presented in Table 6. The diameters of the biscuits were between 55 and 62 mm, representing a 10–24% increase compared to the raw dough (50 mm) during baking. Their height varied between 8 and 10 mm, thus increasing by 14–43% during baking. Based on these values, it was determined that the area of the final products was between 23.98 and 29.71 cm^2^, while their volume was between 19.53 and 26.21 cm^3^. The density of biscuits varied between 0.47 and 0.67 g/cm^3^. The determined baking loss values were between 11.76 and 16.13%. Based on the results, none of the biscuits were significantly different from the control regarding the tested geometric attributes.

### 3.5. Texture Parameters

The results of the texture analysis are presented in Table 7. Values were obtained for hardness (196–875 g), adhesive force (64–183 g), quantity of fractures (5–8), fracturability (121–294 g), cohesiveness (0.12–0.16), springiness (1.39–3.49 mm), gumminess (27–99 g) and chewiness (0.39–2.96 mJ). The results indicate that biscuits containing sunflower pollen at the 10% substitution level possessed significantly lower values for hardness compared to the control. The gumminess and chewiness of the biscuits are determined by hardness; therefore, the changes in these parameters were similar. These changes may be explained by the fact that the addition of pollen decreases the concentration of wheat flour, and thus the gluten content of biscuits. Additionally, the starch composition of the biscuits is also affected by pollen substitution. The 10% rapeseed pollen-containing biscuits showed significantly higher cohesiveness values, while the 10% phacelia-containing biscuits exhibited significantly higher springiness values compared to the control. No statistically significant differences were observed between the control and other samples regarding the adhesive force, quantity of fractures, and fracturability.

### 3.6. Sensory Attributes

#### 3.6.1. Sensory Profile of Biscuits

The MAM-CAP table (Table 8) presents the panel performance. The MAM-CAP table showed that the panel was generally well-trained. Almost all F-Prod values proved to be discriminant (F-Prod *p* < 0.05), except for some odour attributes associated with very small quantities of ingredients (bitter/baking soda/salt odour and flavour). The F-Scal values were adequate (F-Scal *p* > 0.05), with the exception of cut hay, cabbage flavour and odour, brightness, and shape regularity. The sensory panel agreed on all sensory attributes (F-Disag *p* > 0.05). The panel’s repeatability was very good (RMSE ≤ 3.15).

Results of the quantitative descriptive analysis (QDA) are summarized in Table 9. Each biscuit was characterized by regular shape and homogenous surface. Results indicate that the higher the pollen concentration, the more homogenous the surface of the biscuits and the more specks originating from pollen pellets can be seen on them. The lightness of the biscuits showed a similar tendency to that of the obtained spectral colour coordinates: the higher the concentration of a pollen was, the darker the biscuit was. The products containing phacelia pollen were the darkest, followed by those enriched with sunflower and rapeseed pollens. The red colour intensity increased significantly with increasing concentrations of rapeseed and sunflower pollens. The addition of phacelia pollen did not affect the observed red colour intensity of biscuits significantly, although the results of the spectral colour measurement indicated a significant increasement of redness in those samples. The perceived yellow colour intensity was grown by the addition of rapeseed or sunflower pollen, while phacelia pollen caused a significant decrease in this parameter. In contrast to the results of the instrumental colour measurement, the growth was not significant in the cases of sunflower pollen-containing products. The reason for this may be that the red pigments of sunflower pollen mask the yellow colour to the human eye.

The overall odour intensity of biscuits increased, while the flour odour intensity decreased significantly with increasing pollen concentration. Biscuits containing high concentrations of pollen had lower margarine odour intensity. The baking soda odour intensity of the products was low and was not affected by pollen addition. The sweet odour intensity decreased significantly with increasing pollen ratio, probably because the odour of pollen overpowers the sweet odour of biscuits. The mean scores of the sour, bitter and salt odour intensities did not exceed 2.0 in any samples and were not affected by pollen addition. Based on our results, the phacelia pollen-containing biscuits were characterized by a cut hay odour, while rapeseed pollen-containing products possessed a cabbage odour, the intensity of which increased significantly with increasing pollen concentration.

The results showed that the overall flavour intensity of biscuits in which the wheat flour was substituted with 5 or 10% pollen was significantly higher compared to the control. Accordingly, the intensity of the flour flavour decreased significantly with increasing pollen concentration. The mean scores for the sweet taste intensity of the products varied between 43.2 and 55.0, while low mean scores were observed regarding the sour, bitter and salt taste intensities. The sweet taste intensity decreased by rising pollen ratio. Significantly higher bitter and salt taste intensities were observed for biscuits containing pollens at 10% substitution level in comparison to the control. The increase in bitter taste can be attributed to the aroma compounds of the added pollens or to the enhanced Maillard reaction, which is proposed as a pathway of bitter compound formation in thermally processed foods [50]. The results suggest that the cut hay flavour is specific to phacelia pollen, while the cabbage flavour is specific to rapeseed pollen, the intensity of which grow significantly with increasing pollen concentrations in biscuits. Solgajová et al. (2014) also reported that biscuits enriched with rapeseed pollen had a strong cabbage taste, leaving bitterness and spiciness on the tongue [21]. The reason for this observation may be that rapeseed (*Brassica napus* L.) belongs to the *Brassicaceae* family, the members of which usually contain glucosinolates and their hydrolysis products, particularly isothiocyanates [51]. These compounds are responsible for the bitter taste, sulfurous aroma, and pungency of vegetables from the *Brassicaceae* family [52], and presumably for the distinctive flavour of rapeseed pollen-containing biscuits.

The hardness and adhesiveness of biscuits decreased, while chewiness increased as the added pollen ratio rose, regardless of the pollen type. Mouthcoating effect was more characteristic of pollen-containing biscuits in comparison to the control. Biscuits containing pollens at a 10% substitution level had particularly strong mouthcoating effect. The crumbliness and fracturability of biscuits fell significantly with increasing pollen concentration. Most sensory results are not in accordance with the instrumental texture analysis, which can be explained by the fact that the uncertainty is greater for the instrumental measurement due to the heterogeneity of biscuits.

#### 3.6.2. Consumers’ Liking of Biscuits

Consumer liking data were calculated for all consumers and per clusters of consumers. 16%, 39% and 45% of consumers belonged to cluster 1, cluster 2 and cluster 3, respectively. Means ± standard deviations of liking scores are presented in Table 10. Analysing results for all consumers together, significant differences were found in each attribute, which suggests that samples caused varying responses to consumers. Based on the results, consumers preferred the colour of biscuits enriched with rapeseed or sunflower pollens compared to phacelia, especially at higher pollen concentrations. Increasing pollen concentration resulted in a decrease in hardness liking of biscuits within each pollen type; however, the mean scores obtained for the control sample were significantly lower compared to biscuits containing 2% rapeseed or 2% sunflower pollen. None of the samples were significantly different from the control regarding crumbliness liking. The overall/sweet/margarine odour liking, the overall/sweet/margarine flavour liking, and the overall liking scores of biscuits showed similar tendencies. Increasing pollen concentration generally resulted in decreasing liking scores for these attributes. In comparison to the control, significantly lower scores of overall liking were obtained only for biscuits substituted with rapeseed or phacelia pollen at 10% level. In general, biscuits enriched with sunflower pollen were preferred by consumers to other samples. The sensory properties of sunflower pollen-containing biscuits were acceptable even at a 10% concentration level, so it is recommended to use monofloral sunflower pollen for product development.

Comparing results for liking scores between the three clusters, it can be stated that their preferences showed significant differences. Consumers of cluster 1 gave similar scores to each biscuit, except for the 10% phacelia pollen-containing biscuit, which was less preferred by them based on the colour and overall liking scores. Consumers of cluster 2 gave significantly lower liking scores for biscuits containing phacelia pollen at 10% concentration level compared to other products. Additionally, they gave lower mean scores for the control sample in comparison to other clusters. An important characteristic of cluster 3 is that they typically gave lower scores for biscuits containing any type of pollen at 10% concentration level. Based on the overall liking scores, consumers of cluster 1 preferred the 2% phacelia pollen-containing biscuit and the 10% rapeseed pollen-containing biscuit. On the other hand, the product enriched with 2% sunflower pollen was preferred by consumers of the other two clusters the most. The reason for consumers’ preferences may be that biscuits enriched with rapeseed or phacelia pollens were characterized by specific “cabbage” or “cut hay” odour and flavour, while the addition of sunflower pollen did not cause any off-flavours in the products.

#### 3.6.3. Penalty Analysis

Penalty analysis was used to gain understanding of the attributes that most affected the overall liking ratings. Appendix A shows mean drop analysis for the control and pollen-enriched biscuits. Those attributes in the figure that are highlighted suggest that the mean drop and overall penalty are statistically significant, and the product has to be modified in the appropriate direction. The preferred attributes are located at the lower left quadrant of a plot, while attributes of non-optimal levels are located in the upper right quadrant [53]. The control sample was penalized by consumers, since it was “too hard” and it had “not enough global taste”. Consumers considered all samples containing pollen at 5 or 10% substitution level as “too much global taste”. Phacelia or sunflower pollen-containing biscuits appeared to be “too hard” for consumers, but the addition of rapeseed pollen did not result in this effect. As biscuits containing bee pollens at 10% substitution level have the most favourable nutritional properties, it is suggested to focus on these products during product development. These products were in general characterized by “too much global taste”, “too much global flavour”, “not enough sweet taste” and “not enough sweet odour”. Consequently, these inadequacies should be eliminated during future product development.

#### 3.6.4. External Preference Map

An external preference map was created to illustrate the relationship between consumer preference and sensory profile of biscuits, which is presented in Figure 2. Principal component analysis (PCA) was used to visualize the connection between the sensory attributes and samples. Results are presented in a two-dimensional factor plane which explained 73.56% of data variability (F1 = 56.25%; F2 = 17.31%). In the preference map, consumer clusters are modelled as vectors in order to visualize their preferences. The direction of the vector shows which products were preferred by the consumers included in the given cluster, while the length of the vector shows the strength of the preference. In the contour plot, higher consumer acceptance is represented by warmer colours. From the data, three clusters of consumers were obtained from hierarchical clustering analysis. In general, the overall acceptability was oriented towards products containing lower concentrations of pollens, whereas biscuits substituted with pollens at 10% substitution level were the least preferred by consumers. However, consumers of cluster 1 appeared to prefer also those biscuits which contained higher concentrations of rapeseed pollen.

#### 3.6.5. Consumers’ Purchase Intention

Purchase intention was evaluated for all consumers. The results are presented in Table 11. Based on the obtained data, both pollen concentration and pollen type affected the purchase intention of biscuits. It appeared that rising pollen ratio resulted in a decrease in purchase intention; however, biscuits enriched with 2% pollen were more likely to be bought than the control biscuit. Considering results for all consumers together, it can be concluded that consumers were more likely to choose biscuits enriched with sunflower pollen compared to biscuits containing pollen of rapeseed or phacelia.

## 4. Conclusions

In this work, biscuits enriched with rapeseed, phacelia and sunflower pollen at substitution levels of 2, 5 or 10% were compared, based on their nutritional, physical and sensory properties. The results confirmed that pollens are applicable for improving the nutritional value of biscuits; however, further research is needed to understand the interactions between phenolic compounds and macronutrients in pollen enriched biscuits, and to improve the bioaccessibility of nutrients in these products. Consumer acceptance of biscuits was strongly influenced by the botanical origin and concentration level of pollen used for enrichment. Based on the consumer preference test, biscuits enriched with sunflower pollen had more acceptable sensory properties compared to other samples. The reason behind consumers’ preferences may be that biscuits enriched with rapeseed or phacelia pollens were characterized by specific “cabbage” or “cut hay” odours and flavours, while biscuits enriched with sunflower pollen did not have any off-flavours according to the results of sensory profile analysis. This study is the first to demonstrate that pollens of different plant species have heterogeneous effects on the nutritional properties, colour parameters, texture, sensory profile and consumer preference of biscuits. Based on the results of the present paper, it is recommended for food product developers and researchers studying the potential of pollen as a functional food ingredient to use sunflower pollen for the enrichment of foods.

## Figures and Tables

**Figure 1 foods-12-00018-f001:**
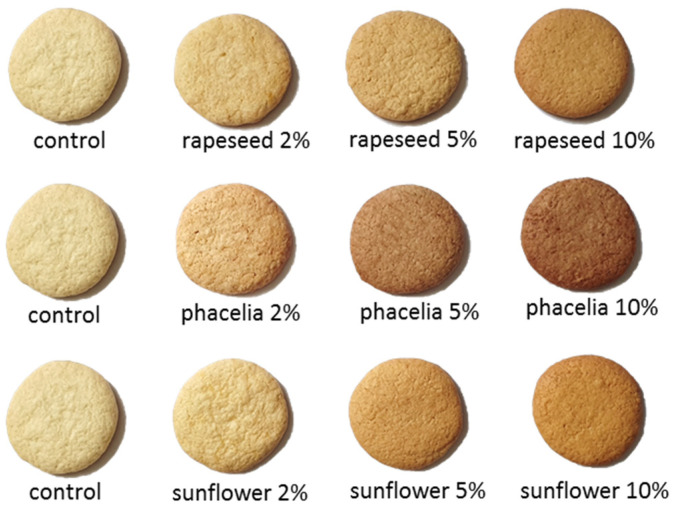
Biscuit samples after baking.

**Figure 2 foods-12-00018-f002:**
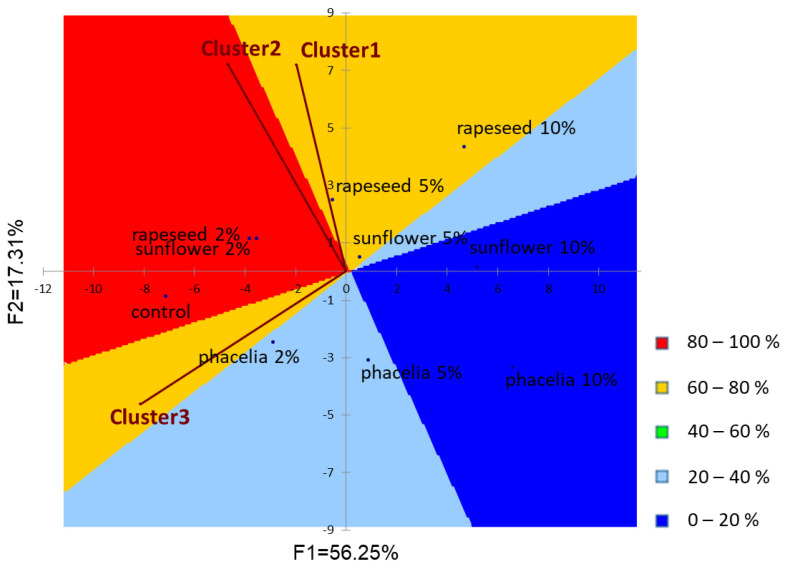
External preference map (vector model) of consumer clusters and biscuits. Higher consumer acceptance is represented by warmer colours.

**Table 1 foods-12-00018-t001:** Raw materials used for biscuit preparation.

Ingredient	Amount (g) (Control Sample)	Amount (g) (Enriched Samples)		
		2% Pollen	5% Pollen	10% Pollen
wheat flour	100.00	98.00	95.00	90.00
pollen	0.00	2.00	5.00	10.00
ground sugar	57.80	57.80	57.80	57.80
margarine	28.40	28.40	28.40	28.40
salt	0.93	0.93	0.93	0.93
distilled water	7.11	7.11	7.11	7.11
glucose syrup	14.60	14.60	14.60	14.60
baking soda	1.11	1.11	1.11	1.11

**Table 2 foods-12-00018-t002:** Macronutrient composition of pollen enriched biscuits (%).

Biscuit	Moisture	Carbohydrate	Crude Protein	Crude Fat	Ash
	(g/100 g Sample)				
control	4.52 ± 0.12 ^a^	77.66 ± 0.47 ^abc^	5.43 ± 0.13 ^c^	11.35 ± 0.27 ^ab^	1.05 ± 0.12 ^b^
rapeseed 2%	4.48 ± 0.09 ^a^	77.64 ± 0.49 ^abc^	5.58 ± 0.19 ^abc^	11.21 ± 0.58 ^ab^	1.09 ± 0.05 ^b^
rapeseed 5%	4.41 ± 0.14 ^ab^	77.10 ± 0.17 ^abc^	5.74 ± 0.08 ^abc^	11.66 ± 0.37 ^ab^	1.09 ± 0.15 ^b^
rapeseed 10%	4.33 ± 0.14 ^abc^	76.08 ± 0.39 ^bc^	6.49 ± 0.30 ^ab^	11.72 ± 0.17 ^ab^	1.38 ± 0.04 ^a^
phacelia 2%	4.17 ± 0.13 ^abc^	77.03 ± 0.63 ^abc^	5.54 ± 0.24 ^abc^	12.18 ± 0.35 ^a^	1.08 ± 0.01 ^ab^
phacelia 5%	3.94 ± 0.05 ^c^	75.99 ± 0.67 ^c^	6.08 ± 0.20 ^abc^	12.83 ± 0.50 ^a^	1.16 ± 0.02 ^ab^
phacelia 10%	4.01 ± 0.12 ^bc^	76.24 ± 0.73 ^bc^	6.56 ± 0.09 ^a^	12.06 ± 0.65 ^ab^	1.13 ± 0.07 ^ab^
sunflower 2%	4.37 ± 0.13 ^ab^	78.06 ± 0.39 ^ab^	5.56 ± 0.22 ^abc^	10.91 ± 0.40 ^b^	1.10 ± 0.04 ^ab^
sunflower 5%	4.22 ± 0.16 ^abc^	78.45 ± 0.61 ^a^	5.46 ± 0.24 ^bc^	10.70 ± 0.66 ^b^	1.18 ± 0.10 ^ab^
sunflower 10%	4.05 ± 0.13 ^abc^	77.32 ± 0.43 ^abc^	5.44 ± 0.18 ^c^	12.08 ± 0.40 ^ab^	1.10 ± 0.08 ^ab^

Data are presented as mean ± standard deviation (*n* = 4). Different letters represent significant difference between means within each column (*p* < 0.05; Kruskal–Wallis test, Dunn’s pairwise procedure with Bonferroni correction).

**Table 3 foods-12-00018-t003:** Total phenolic content (TPC) and antioxidant capacity (FRAP, TEAC, DPPH) of pollen enriched biscuits.

Biscuit	TPC (mg GAE/100 g)	FRAP (mg AAE/100 g)	TEAC (mg TE/100 g)	DPPH (mg TE/100 g)
control	50.53 ± 1.61 ^d^	13.42 ± 1.46 ^d^	21.17 ± 1.68 ^c^	11.66 ± 2.75 ^c^
rapeseed 2%	57.54 ± 2.00 ^cd^	18.45 ± 0.86 ^cd^	22.77 ± 1.14 ^bc^	24.33 ± 2.77 ^bc^
rapeseed 5%	71.49 ± 1.29 ^abcd^	32.42 ± 2.28 ^abcd^	31.61 ± 1.21 ^abc^	49.74 ± 3.04 ^abc^
rapeseed 10%	98.13 ± 3.02 ^ab^	46.51 ± 2.51 ^abc^	42.18 ± 1.61 ^ab^	68.52 ± 6.31 ^ab^
phacelia 2%	65.63 ± 1.62 ^abcd^	21.86 ± 1.69 ^bcd^	27.14 ± 1.24 ^abc^	36.97 ± 1.82 ^abc^
phacelia 5%	90.83 ± 5.39 ^abc^	41.47 ± 2.64 ^abcd^	36.72 ± 0.94 ^abc^	75.51 ± 1.16 ^a^
phacelia 10%	123.66 ± 2.15 ^a^	63.54 ± 2.16 ^a^	52.92 ± 0.91 ^a^	104.25 ± 1.53 ^a^
sunflower 2%	60.22 ± 1.36 ^cd^	23.25 ± 0.82 ^bcd^	21.42 ± 1.74 ^c^	26.54 ± 6.49 ^bc^
sunflower 5%	75.88 ± 2.65 ^abcd^	37.57 ± 2.03 ^abcd^	34.63 ± 0.96 ^abc^	58.35 ± 1.90 ^abc^
sunflower 10%	93.46 ± 1.14 ^abc^	57.81 ± 1.64 ^ab^	42.60 ± 1.13 ^a^	70.51 ± 2.39 ^ab^

Data are presented as mean ± standard deviation (*n* = 4). Different letters represent significant difference between means within each column (*p* < 0.05; Kruskal–Wallis test, Dunn’s pairwise procedure with Bonferroni correction).

**Table 4 foods-12-00018-t004:** CIELAB colour coordinates, chroma and hue angle of biscuits.

Biscuit	L*	a*	b*	C*	h°
control	70.3 ± 0.72 ^a^	1.0 ± 0.48 ^f^	31.8 ± 0.58 ^f^	31.8 ± 0.59 ^g^	88.2 ± 0.84 ^a^
rapeseed 2%	67.4 ± 0.94 ^b^	4.5 ± 0.63 ^e^	32.9 ± 0.32 ^e^	33.2 ± 0.37 ^f^	82.3 ± 1.04 ^b^
rapeseed 5%	64.4 ± 1.01 ^c^	5.9 ± 0.56 ^d^	34.2 ± 0.34 ^d^	34.7 ± 0.40 ^e^	80.2 ± 0.86 ^d^
rapeseed 10%	60.9 ± 1.38 ^d^	7.3 ± 0.70 ^bc^	35.3 ± 0.34 ^c^	36.1 ± 0.31 ^c^	78.3 ± 1.13 ^e^
phacelia 2%	61.6 ± 0.74 ^d^	4.6 ± 0.52 ^e^	29.6 ± 0.39 ^g^	30.0 ± 0.44 ^h^	81.2 ± 0.92 ^c^
phacelia 5%	55.9 ± 1.36 ^f^	6.9 ± 0.43 ^c^	27.6 ± 0.47 ^h^	28.5 ± 0.55 ^i^	75.9 ± 0.65 ^f^
phacelia 10%	50.1 ± 1.36 ^g^	9.1 ± 0.47 ^a^	26.7 ± 0.67 ^i^	28.2 ± 0.68 ^i^	71.2 ± 0.92 ^g^
sunflower 2%	68.0 ± 0.87 ^b^	4.6 ± 0.57 ^e^	34.9 ± 0.76 ^c^	35.2 ± 0.73 ^d^	82.5 ± 0.98 ^b^
sunflower 5%	61.3 ± 0.98 ^d^	7.9 ± 0.47 ^b^	36.8 ± 0.53 ^b^	37.7 ± 0.54 ^b^	77.8 ± 0.70 ^e^
sunflower 10%	59.5 ± 0.89 ^e^	8.8 ± 0.68 ^a^	40.5 ± 0.81 ^a^	41.5 ± 0.86 ^a^	77.7 ± 0.87 ^e^

Data are presented as mean ± standard deviation (*n* = 12). Different letters represent significant difference between means within each column (*p* < 0.05; ANOVA test, Tukey-HSD post hoc test). L*: lightness from black (0) to white (100); a*: red–green colour (a* > 0 indicates redness, a* < 0 indicates greenness); b*: yellow–blue colour (b* > 0 indicates yellowness, b* < 0 indicates blueness).

**Table 5 foods-12-00018-t005:** Colour difference (ΔE_ab*_) values of biscuits.

Biscuit	Control	Rapeseed 2%	Rapeseed 5%	Rapeseed 10%	Phacelia 2%	Phacelia 5%	Phacelia 10%	Sunflower 2%	Sunflower 5%	Sunflower 10%
control		4.6	8.0	11.9	9.6	16.1	22.3	5.3	12.4	15.9
rapeseed 2%			3.6	7.5	6.7	12.9	19.0	2.1	8.1	11.8
rapeseed 5%				3.9	5.5	10.8	16.4	3.9	4.6	8.5
rapeseed 10%					6.4	9.2	13.9	7.6	1.7	5.6
phacelia 2%						6.5	12.7	8.3	8.0	11.9
phacelia 5%							6.3	14.3	10.7	13.5
phacelia 10%								20.2	15.1	16.7
sunflower 2%									7.8	11.1
sunflower 5%										4.2
sunflower 10%										

0 < ΔE_ab*_ < 1—observer does not notice the difference. 1 < ΔE_ab*_ < 2—only an experienced observer can notice the difference. 2 < ΔE_ab*_ < 3:5—an unexperienced observer also notices the difference. 3.5 < ΔE_ab*_ < 5—clear difference in colour is noticed. 5 < ΔE_ab*_—observer notices two different colours [49].

**Table 6 foods-12-00018-t006:** Geometric properties and baking loss of biscuits.

Biscuit	Diameter (mm)	Height (mm)	Volume (cm^3^)	Area (cm^2^)	Density (g/cm^3^)	Baking Loss (%)
control	60 ± 0.96 ^ab^	9 ± 0.96 ^ab^	24.60 ± 3.51 ^ab^	28.04 ± 0.90 ^ab^	0.53 ± 0.10 ^ab^	14.78 ± 1.60 ^ab^
rapeseed 2%	59 ± 2.22 ^ab^	9 ± 0.50 ^ab^	23.76 ± 2.50 ^ab^	27.14 ± 2.04 ^ab^	0.54 ± 0.05 ^ab^	13.92 ± 1.65 ^ab^
rapeseed 5%	57 ± 1.71 ^ab^	8 ± 0.00 ^ab^	20.25 ± 1.22 ^ab^	25.31 ± 1.53 ^ab^	0.64 ± 0.04 ^ab^	15.58 ± 3.33 ^ab^
rapeseed 10%	58 ± 1.29 ^ab^	8 ± 0.50 ^a^	20.14 ± 1.77 ^ab^	25.30 ± 0.86 ^ab^	0.67 ± 0.06 ^b^	12.94 ± 0.18 ^a^
phacelia 2%	55 ± 0.50 ^a^	9 ± 0.58 ^ab^	20.39 ± 1.63 ^ab^	23.98 ± 0.44 ^a^	0.63 ± 0.07 ^ab^	15.72 ± 0.32 ^b^
phacelia 5%	56 ± 0.96 ^a^	8 ± 0.00 ^ab^	19.53 ± 0.67 ^a^	24.42 ± 0.84 ^a^	0.64 ± 0.02 ^ab^	13.76 ± 0.36 ^ab^
phacelia 10%	58 ± 0.50 ^ab^	8 ± 0.00 ^ab^	20.96 ± 0.36 ^ab^	26.20 ± 0.45 ^ab^	0.60 ± 0.02 ^ab^	16.13 ± 0.60 ^b^
sunflower 2%	59 ± 0.96 ^ab^	10 ± 0.58 ^b^	26.21 ± 2.00 ^b^	26.90 ± 1.75 ^ab^	0.49 ± 0.05 ^ab^	14.68 ± 0.51 ^ab^
sunflower 5%	62 ± 0.58 ^b^	9 ± 0.50 ^ab^	25.98 ± 1.29 ^b^	29.71 ± 0.56 ^b^	0.47 ± 0.04 ^a^	13.79 ± 0.53 ^ab^
sunflower 10%	59 ± 1.41 ^ab^	8 ± 0.50 ^ab^	22.59 ± 2.11 ^ab^	26.96 ± 1.56 ^ab^	0.55 ± 0.05 ^ab^	14.70 ± 0.40 ^ab^

Data are presented as mean ± standard deviation (*n* = 4). Different letters represent significant difference between means within each column (*p* < 0.05; Kruskal–Wallis test, Dunn’s pairwise procedure with Bonferroni correction).

**Table 7 foods-12-00018-t007:** Texture properties of biscuits.

Biscuit	Hardness (g)	Adhesive Force (g)	Fracturability (g)	Quantity of Fractures	Cohesiveness	Springiness (mm)	Guminess (g)	Chewiness (mJ)
control	454 ± 108.14 ^bcd^	80 ± 14.67 ^abc^	170 ± 80.75 ^abc^	7 ± 1.58 ^a^	0.13 ± 0.01 ^a^	1.36 ± 0.14 ^ab^	64 ± 17.89 ^bcd^	1.05 ± 0.23 ^bcde^
rapeseed 2%	472 ± 114.78 ^bcd^	131 ± 24.16 ^c^	176 ± 25.37 ^abc^	8 ± 2.05 ^a^	0.13 ± 0.01 ^a^	2.22 ± 0.58 ^bc^	66 ± 10.22 ^bcd^	1.40 ± 0.33 ^cde^
rapeseed 5%	713 ± 107.96 ^d^	130 ± 56.80 ^abc^	263 ± 121.44 ^bc^	7 ± 1.73 ^a^	0.14 ± 0.01 ^ab^	2.08 ± 0.95 ^abc^	99 ± 9.70 ^d^	2.18 ± 0.87 ^de^
rapeseed 10%	269 ± 93.21 ^ab^	66 ± 30.64 ^ab^	121 ± 10.41 ^a^	6 ± 1.48 ^a^	0.16 ± 0.03 ^b^	1.88 ± 0.55 ^ab^	41 ± 9.29 ^abc^	0.87 ± 0.43 ^abcd^
phacelia 2%	620 ± 129.13 ^cd^	183 ± 78.22 ^c^	268 ± 81.45 ^bc^	7 ± 1.32 ^a^	0.13 ± 0.02 ^ab^	1.41 ± 0.18 ^ab^	72 ± 11.37 ^cd^	1.07 ± 0.34 ^bcde^
phacelia 5%	242 ± 86.61 ^ab^	89 ± 28.48 ^abc^	154 ± 27.62 ^ab^	6 ± 2.03 ^a^	0.14 ± 0.01 ^ab^	1.52 ± 0.25 ^abc^	30 ± 4.45 ^ab^	0.55 ± 0.16 ^ab^
phacelia 10%	875 ± 235.37 ^d^	136 ± 43.62 ^bc^	294 ± 43.30 ^c^	7 ± 1.00 ^a^	0.12 ± 0.01 ^a^	3.49 ± 0.83 ^c^	105 ± 12.14 ^d^	2.96 ± 0.60 ^e^
sunflower 2%	283 ± 78.64 ^abc^	91 ± 30.25 ^abc^	190 ± 38.17 ^abc^	6 ± 1.73 ^a^	0.13 ± 0.01 ^ab^	1.39 ± 0.30 ^a^	34 ± 7.44 ^abc^	0.52 ± 0.10 ^ab^
sunflower 5%	258 ± 11.99 ^abc^	68 ± 27.18 ^ab^	168 ± 48.81 ^abc^	6 ± 0.71 ^a^	0.13 ± 0.02 ^ab^	1.60 ± 0.28 ^abc^	30 ± 5.41 ^ab^	0.58 ± 0.14 ^abc^
sunflower 10%	196 ± 38.25 ^a^	64 ± 12.21 ^a^	155 ± 16.50 ^ab^	5 ± 1.50 ^a^	0.14 ± 0.01 ^ab^	1.49 ± 0.14 ^ab^	27 ± 5.07 ^a^	0.39 ± 0.08 ^a^

Data are presented as mean ± standard deviation (*n* = 9). Different letters represent significant difference between means within each column (*p* < 0.05; Kruskal–Wallis test, Dunn’s pairwise procedure with Bonferroni correction).

**Table 8 foods-12-00018-t008:** Sensory panel performance MAM-CAP table.

Attribute	F-Prod	F-Scal	F-Disag	RMSE
yellow colour intensity	1571.13	0.92	0.85	2.54
red colour intensity	1232.00	0.96	0.92	1.66
cabbage flavour intensity	1141.42	2.04 *	0.97	1.43
cut hay flavour intensity	1128.14	3.27 *	0.54	1.29
cut hay odour intensity	1003.56	2.21 *	0.71	1.43
cabbage odour intensity	877.93	4.05 *	0.56	1.96
global odour intensity	641.21	1.81	0.76	2.37
global flavour intensity	386.40	1.40	0.84	3.15
flour odour intensity	340.47	0.87	1.11	2.35
amount of specks on the surface	289.45	0.32	0.93	0.96
flour flavour intensity	179.94	0.69	1.08	1.44
adhesiveness	96.47	1.39	1.07	1.46
fracturability	67.69	0.80	1.02	2.53
sweet taste intensity	48.50	1.46	1.08	2.48
sweet odour intensity	37.25	0.78	1.08	2.90
mouthcoating	29.30	0.37	1.17	1.87
hardness	27.99	0.80	1.08	2.21
chewiness	21.99	1.36	0.83	2.03
crumbliness	21.63	0.57	1.22	1.48
margarine odour intensity	14.14	0.83	0.64	2.86
brightness	9.62	4.80 *	0.69	46.88 *
bitter taste intensity	6.99	0.63	0.97	1.48
margarine flavour intensity	6.24	0.71	1.33	1.19
salt taste intensity	5.90	0.83	0.77	1.29
shape regularity	5.84	2.08 *	0.91	0.79
bitter odour intensity	1.39 *	0.98	0.79	1.76
baking soda flavour intensity	1.38 *	0.61	0.54	1.81
salt odour intensity	0.98 *	1.55	0.63	1.20
sour taste intensity	0.76 *	1.78	0.90	1.31
baking soda odour intensity	0.60 *	0.35	0.93	1.85
sour odour intensity	0.26 *	0.31	1.14	1.71

F statistics of discrimination (F-Prod), scaling heterogeneity (F-Scal), disagreement (F-Disag), and repeatability (root mean squares of error, RMSE). Significant values are marked with an asterisk (*) (*F*-Prod *p* < 0.05; *F*-Scal *p* > 0.05; *F*-Disag *p* > 0.05; RMSE ≤ 3.15).

**Table 9 foods-12-00018-t009:** Sensory properties of biscuits (quantitative descriptive analysis, QDA).

Sensory Attributes	Biscuit									
	Control	Rapeseed 2%	Rapeseed 5%	Rapeseed 10%	Phacelia 2%	Phacelia 5%	Phacelia 10%	Sunflower 2%	Sunflower 5%	Sunflower 10%
shape regularity	95.0 ± 0.9 ^a^	94.3 ± 0.7 ^b^	94.1 ± 0.9 ^b^	93.9 ± 0.7 ^b^	94.1 ± 0.8 ^b^	94.2 ± 0.6 ^b^	94.0 ± 0.7 ^b^	93.7 ± 0.8 ^b^	93.8 ± 0.8 ^b^	93.9 ± 0.8 ^b^
homogeneity of the surface	81.8 ± 2.1 ^d^	82.5 ± 1.7 ^d^	83.0 ± 1.7 ^d^	91.1 ± 2.8 ^b^	82.4 ± 1.4 ^d^	93.9 ± 1.2 ^a^	94.5 ± 1.3 ^a^	82.0 ± 1.6 ^d^	85.8 ± 1.4 ^c^	93.7 ± 1.6 ^a^
amount of specks on the surface	0.0 ± 0.0 ^d^	2.3 ± 1.2 ^c^	4.4 ± 0.9 ^b^	9.1 ± 1.2 ^a^	3.0 ± 0.8 ^c^	4.6 ± 0.9 ^b^	9.3 ± 0.9 ^a^	2.6 ± 0.9 ^c^	4.5 ± 0.8 ^b^	8.9 ± 1.0 ^a^
lightness	88.3 ± 2.2 ^a^	79.2 ± 1.7 ^b^	75.1 ± 1.1 ^c^	68.4 ± 2.1 ^d^	49.7 ± 2.2 ^f^	38.7 ± 2.5 ^g^	24.5 ± 2.3 ^h^	80.3 ± 1.4 ^b^	67.1 ± 2.0 ^d^	60.8 ± 2.6 ^e^
red colour intensity	4.5 ± 1.4 ^f^	5.0 ± 1.4 ^ef^	6.1 ± 1.6 ^de^	7.5 ± 2.0 ^cd^	5.1 ± 1.3 ^ef^	6.0 ± 1.3 ^ef^	6.0 ± 1.3 ^ef^	8.5 ± 1.5 ^c^	23.8 ± 2.2 ^b^	39.2 ± 2.0 ^a^
yellow colour intensity	40.9 ± 2.3 ^d^	51.4 ± 2.2 ^c^	61.9 ± 2.0 ^b^	80.9 ± 4.5 ^a^	31.5 ± 2.9 ^e^	24.2 ± 1.6 ^f^	15.7 ± 1.1 ^g^	50.6 ± 2.0 ^c^	51.4 ± 2.3 ^c^	52.7 ± 2.9 ^c^
overall odour intensity	54.2 ± 2.6 ^d^	60.5 ± 2.0 ^c^	69.5 ± 2.3 ^b^	82.9 ± 2.1 ^a^	60.2 ± 1.9 ^c^	70.2 ± 2.7 ^b^	82.3 ± 2.3 ^a^	59.2 ± 1.9 ^c^	70.5 ± 2.8 ^b^	82.5 ± 2.2 ^a^
flour odour intensity	59.8 ± 2.3 ^a^	55.5 ± 2.3 ^b^	51.6 ± 2.6 ^c^	42.7 ± 2.2 ^d^	54.5 ± 2.5 ^b^	51.1 ± 2.4 ^c^	32.5 ± 2.2 ^e^	55.7 ± 2.6 ^b^	50.9 ± 2.3 ^c^	33.9 ± 2.5 ^e^
margarine odour intensity	21.8 ± 2.4 ^a^	21.3 ± 2.6 ^ab^	22.2 ± 2.3 ^a^	19.1 ± 3.3 ^c^	21.7 ± 2.6 ^a^	19.8 ± 2.6 ^bc^	17.0 ± 1.9 ^d^	22.0 ± 2.1 ^a^	19.1 ± 1.8 ^c^	18.7 ± 3.5 ^c^
baking soda odour intensity	1.6 ± 1.9 ^a^	2.0 ± 2.0 ^a^	1.5 ± 1.7 ^a^	1.7 ± 2.0 ^a^	1.8 ± 2.0 ^a^	1.6 ± 1.6 ^a^	1.6 ± 1.2 ^a^	1.0 ± 1.6 ^a^	1.9 ± 1.9 ^a^	1.5 ± 1.8 ^a^
sweet odour intensity	53.8 ± 2.9 ^a^	50.5 ± 3.1 ^bc^	44.4 ± 2.8 ^d^	43.9 ± 2.8 ^d^	52.0 ± 2.2 ^ab^	50.0 ± 3.4 ^bc^	44.2 ± 2.9 ^d^	49.0 ± 3.1 ^c^	45.7 ± 3.2 ^d^	43.7 ± 2.9 ^d^
sour odour intensity	1.8 ± 1.6 ^a^	1.5 ± 1.8 ^a^	1.6 ± 1.6 ^a^	1.6 ± 2.0 ^a^	1.8 ± 1.8 ^a^	1.6 ± 1.5 ^a^	1.7 ± 1.5 ^a^	1.3 ± 1.6 ^a^	1.6 ± 1.8 ^a^	2.0 ± 2.1 ^a^
bitter odour intensity	0.4 ± 1.0 ^a^	0.5 ± 1.2 ^a^	0.8 ± 1.5 ^a^	1.1 ± 1.9 ^a^	0.6 ± 1.3 ^a^	1.2 ± 2.1 ^a^	1.5 ± 2.3 ^a^	0.8 ± 1.4 ^a^	1.0 ± 1.7 ^a^	1.4 ± 2.2 ^a^
salt odour intensity	0.4 ± 0.9 ^a^	0.4 ± 0.8 ^a^	0.6 ± 1.1 ^a^	0.7 ± 1.3 ^a^	0.7 ± 1.3 ^a^	0.8 ± 1.5 ^a^	0.8 ± 1.5 ^a^	0.3 ± 0.7 ^a^	0.5 ± 1.1 ^a^	0.5 ± 1.3 ^a^
cut hay odour intensity	0.3 ± 0.8 ^d^	0.3 ± 0.8 ^d^	0.5 ± 0.9 ^d^	0.7 ± 1.3 ^d^	10.4 ± 1.6 ^c^	15.5 ± 2.2 ^b^	21.3 ± 2.3 ^a^	0.5 ± 0.9 ^d^	0.5 ± 1.0 ^d^	0.5 ± 1.1 ^d^
cabbage odour intensity	0.0 ± 0.0 ^d^	9.0 ± 1.7 ^c^	17.6 ± 3.0 ^b^	24.9 ± 3.1 ^a^	0.6 ± 1.4 ^d^	0.8 ± 1.7 ^d^	1.0 ± 1.8 ^d^	0.3 ± 0.9 ^d^	0.4 ± 1.1 ^d^	0.6 ± 1.5 ^d^
overall flavour intensity	59.9 ± 3.0 ^d^	60.0 ± 3.8 ^d^	73.4 ± 3.0 ^c^	84.0 ± 2.3 ^b^	60.6 ± 3.6 ^d^	72.4 ± 2.9 ^c^	90.9 ± 2.8 ^a^	61.0 ± 3.5 ^d^	71.2 ± 2.9 ^c^	84.8 ± 2.6 ^b^
flour flavour intensity	17.4 ± 2.1 ^a^	12.7 ± 1.8 ^b^	9.3 ± 1.6 ^c^	4.8 ± 0.8 ^d^	12.2 ± 1.6 ^b^	10.0 ± 1.2 ^c^	4.8 ± 1.0 ^d^	11.9 ± 1.6 ^b^	10.0 ± 1.6 ^c^	4.9 ± 0.7 ^d^
margarine flavour intensity	10.5 ± 1.2 ^ab^	10.8 ± 1.4 ^a^	10.7 ± 1.1 ^ab^	9.8 ± 1.4 ^abc^	10.0 ± 1.3 ^abc^	9.5 ± 1.7 ^bc^	9.0 ± 1.0 ^c^	10.5 ± 1.4 ^ab^	9.0 ± 0.9 ^c^	9.2 ± 1.2 ^c^
baking soda flavour intensity	1.8 ± 1.8 ^a^	1.3 ± 1.5 ^a^	1.3 ± 1.6 ^a^	2.0 ± 2.1 ^a^	1.3 ± 1.6 ^a^	1.1 ± 1.2 ^a^	1.2 ± 1.4 ^a^	1.1 ± 1.4 ^a^	1.7 ± 1.7 ^a^	1.0 ± 1.4 ^a^
sweet taste intensity	55.0 ± 1.8 ^a^	50.2 ± 2.9 ^b^	44.0 ± 2.4 ^cd^	43.2 ± 2.3 ^d^	48.2 ± 3.3 ^b^	48.9 ± 1.3 ^b^	45.5 ± 3.8 ^c^	49.5 ± 2.4 ^b^	45.1 ± 2.3 ^cd^	44.1 ± 2.3 ^cd^
sour taste intensity	1.3 ± 1.2 ^a^	1.2 ± 1.4 ^a^	1.4 ± 1.3 ^a^	0.9 ± 1.2 ^a^	1.6 ± 1.4 ^a^	1.1 ± 1.3 ^a^	1.3 ± 1.1 ^a^	0.9 ± 1.1 ^a^	1.3 ± 1.4 ^a^	1.3 ± 1.5 ^a^
bitter taste intensity	0.4 ± 1.0 ^d^	1.3 ± 1.4 ^bcd^	1.6 ± 1.7 ^abcd^	2.5 ± 2.1 ^ab^	1.1 ± 1.1 ^bcd^	1.9 ± 1.5 ^abc^	2.8 ± 1.6 ^a^	0.9 ± 1.1 ^cd^	1.6 ± 1.5 ^abcd^	2.3 ± 1.6 ^ab^
salt taste intensity	1.0 ± 1.0 ^c^	1.0 ± 1.1 ^c^	1.3 ± 1.3 ^bc^	2.3 ± 1.2 ^ab^	1.5 ± 1.4 ^abc^	1.7 ± 1.4 ^abc^	2.4 ± 1.1 ^ab^	1.5 ± 1.4 ^abc^	1.9 ± 1.2 ^abc^	2.5 ± 1.1 ^a^
cut hay flavour intensity	0.4 ± 0.8 ^d^	0.6 ± 0.9 ^d^	0.6 ± 0.9 ^d^	0.6 ± 0.9 ^d^	9.8 ± 1.1 ^c^	13.4 ± 1.8 ^b^	17.1 ± 2.2 ^a^	0.5 ± 0.8 ^d^	0.5 ± 0.9 ^d^	0.5 ± 0.8 ^d^
cabbage flavour intensity	0.0 ± 0.0 ^d^	9.2 ± 1.3 ^c^	18.9 ± 1.7 ^b^	27.6 ± 2.5 ^a^	0.6 ± 0.9 ^d^	0.8 ± 1.2 ^d^	0.9 ± 1.2 ^d^	0.3 ± 0.8 ^d^	0.3 ± 0.8 ^d^	0.7 ± 1.0 ^d^
hardness	33.3 ± 2.5 ^a^	31.8 ± 2.3 ^ab^	28.3 ± 1.8 ^de^	26.1 ± 2.0 ^f^	31.0 ± 2.2 ^bc^	29.0 ± 2.4 ^cd^	26.4 ± 2.7 ^ef^	31.2 ± 2.0 ^b^	29.0 ± 2.2 ^cd^	26.7 ± 1.5 ^ef^
chewiness	22.4 ± 1.8 ^d^	22.3 ± 1.5 ^d^	23.8 ± 2.5 ^cd^	25.6 ± 1.6 ^ab^	22.0 ± 2.3 ^d^	24.4 ± 2.1 ^bc^	26.7 ± 2.3 ^a^	22.2 ± 1.9 ^d^	24.7 ± 1.9 ^bc^	26.2 ± 1.6 ^ab^
adhesiveness	5.7 ± 1.2 ^e^	8.8 ± 0.9 ^cd^	10.1 ± 1.1 ^bc^	15.4 ± 1.7 ^a^	9.1 ± 0.9 ^bcd^	10.0 ± 1.3 ^bcd^	14.6 ± 2.5 ^a^	8.7 ± 1.1 ^d^	10.4 ± 1.1 ^b^	14.0 ± 2.3 ^a^
mouthcoating	16.3 ± 1.2 ^c^	19.1 ± 1.7 ^b^	19.5 ± 1.8 ^b^	23.6 ± 1.6 ^a^	19.0 ± 2.4 ^b^	19.0 ± 1.9 ^b^	22.9 ± 2.5 ^a^	19.6 ± 1.6 ^b^	19.1 ± 1.7 ^b^	22.5 ± 2.1 ^a^
crumbliness	19.5 ± 1.7 ^a^	16.8 ± 1.5 ^bc^	15.8 ± 1.7 ^cd^	14.6 ± 1.8 ^d^	18.0 ± 1.7 ^b^	15.5 ± 1.7 ^cd^	15.1 ± 1.2 ^d^	16.9 ± 1.4 ^bc^	15.3 ± 1.3 ^d^	15.1 ± 1.2 ^d^
fracturability	53.4 ± 2.4 ^a^	46.5 ± 3.1 ^c^	44.5 ± 2.2 ^c^	42.1 ± 2.1 ^d^	49.6 ± 2.0 ^b^	46.6 ± 3.6 ^c^	39.9 ± 2.3 ^d^	49.0 ± 2.2 ^b^	48.9 ± 2.6 ^b^	41.5 ± 1.8 ^d^

Data are presented as mean ± standard deviation (*n* = 24). Different letters represent significant difference between means within each raw (*p* < 0.05; ANOVA test, Tukey-HSD post hoc test).

**Table 10 foods-12-00018-t010:** Consumer liking scores of biscuits.

Sensory Attributes	Consumer Groups	Biscuit									
		Control	Rapeseed 2%	Rapeseed 5%	Rapeseed 10%	Phacelia 2%	Phacelia 5%	Phacelia 10%	Sunflower 2%	Sunflower 5%	Sunflower 10%
colour liking	cluster 1	5.6 ± 2.2 ^abc^	6.5 ± 1.3 ^ab^	6.3 ± 1.9 ^ab^	6.8 ± 1.6 ^ab^	6.4 ± 1.6 ^ab^	4.3 ± 1.8 ^c^	5.1 ± 2.4 ^bc^	7.1 ± 1.1 ^a^	7.3 ± 1.3 ^a^	6.9 ± 1.9 ^ab^
	cluster 2	5.8 ± 2.1 ^bc^	6.1 ± 1.7 ^abc^	7.2 ± 1.6 ^a^	6.7 ± 1.8 ^ab^	5.8 ± 2.1 ^bc^	5.1 ± 1.9 ^cd^	4.1 ± 2.1 ^d^	6.7 ± 1.7 ^ab^	6.7 ± 1.6 ^ab^	6.8 ± 1.7 ^ab^
	cluster 3	6.7 ± 1.5 ^ab^	6.5 ± 1.3 ^ab^	6.9 ± 1.6 ^a^	6.7 ± 1.9 ^ab^	5.6 ± 1.9 ^bc^	4.6 ± 2.1 ^c^	5.0 ± 2.3 ^c^	6.8 ± 1.8 ^ab^	7.0 ± 1.6 ^a^	6.4 ± 2.0 ^ab^
	all consumers	6.2 ± 1.9 ^ab^	6.4 ± 1.4 ^ab^	6.9 ± 1.7 ^a^	6.7 ± 1.8 ^a^	5.8 ± 1.9 ^b^	4.7 ± 2.0 ^c^	4.7 ± 2.3 ^c^	6.8 ± 1.7 ^a^	6.9 ± 1.6 ^a^	6.7 ± 1.9 ^a^
overall odour liking	cluster 1	6.0 ± 1.5 ^a^	5.6 ± 2.2 ^a^	5.1 ± 1.9 ^a^	5.8 ± 2.6 ^a^	6.4 ± 1.4 ^a^	5.6 ± 1.1 ^a^	5.4 ± 2.2 ^a^	5.6 ± 1.5 ^a^	6.8 ± 1.3 ^a^	6.2 ± 1.8 ^a^
	cluster 2	5.6 ± 1.8 ^a^	5.8 ± 1.9 ^a^	6.1 ± 2.0 ^a^	5.3 ± 2.1 ^a^	6.0 ± 1.9 ^a^	5.7 ± 1.7 ^a^	3.9 ± 1.9 ^b^	6.3 ± 1.6 ^a^	5.6 ± 1.7 ^a^	6.2 ± 2.2 ^a^
	cluster 3	6.0 ± 1.5 ^a^	5.6 ± 1.7 ^a^	5.2 ± 2.0 ^ab^	3.7 ± 2.3 ^c^	5.5 ± 1.5 ^ab^	5.5 ± 1.5 ^ab^	4.4 ± 1.9 ^bc^	6.3 ± 1.4 ^a^	6.3 ± 1.5 ^a^	5.3 ± 2.3 ^ab^
	all consumers	5.9 ± 1.6 ^a^	5.7 ± 1.8 ^a^	5.6 ± 2.0 ^a^	4.7 ± 2.4 ^b^	5.8 ± 1.6 ^a^	5.6 ± 1.5 ^a^	4.4 ± 2.0 ^b^	6.2 ± 1.5 ^a^	6.1 ± 1.6 ^a^	5.8 ± 2.2 ^a^
sweet odour liking	cluster 1	5.5 ± 2.1 ^a^	5.4 ± 2.2 ^a^	5.6 ± 1.1 ^a^	5.2 ± 2.2 ^a^	5.4 ± 1.7 ^a^	5.4 ± 1.4 ^a^	5.0 ± 1.5 ^a^	5.3 ± 1.4 ^a^	5.7 ± 1.4 ^a^	5.6 ± 2.0 ^a^
	cluster 2	5.4 ± 1.6 ^a^	5.7 ± 1.6 ^a^	6.2 ± 2.0 ^a^	5.0 ± 2.1 ^ab^	5.6 ± 2.0 ^a^	5.6 ± 1.6 ^a^	3.8 ± 1.7 ^b^	5.6 ± 1.6 ^a^	5.2 ± 1.7 ^a^	5.8 ± 1.9 ^a^
	cluster 3	5.8 ± 2.2 ^a^	5.4 ± 2.1 ^ab^	5.4 ± 2.0 ^ab^	4.0 ± 2.3 ^c^	5.4 ± 1.6 ^ab^	5.1 ± 1.7 ^abc^	4.2 ± 2.0 ^bc^	5.9 ± 1.8 ^a^	6.3 ± 1.7 ^a^	5.1 ± 2.0 ^abc^
	all consumers	5.6 ± 1.9 ^a^	5.5 ± 1.9 ^a^	5.8 ± 1.9 ^a^	4.6 ± 2.2 ^bc^	5.5 ± 1.8 ^a^	5.4 ± 1.6 ^ab^	4.2 ± 1.9 ^c^	5.7 ± 1.7 ^a^	5.8 ± 1.7 ^a^	5.5 ± 2.0 ^a^
margarine odour liking	cluster 1	5.0 ± 1.2 ^a^	5.1 ± 1.8 ^a^	5.8 ± 1.1 ^a^	5.2 ± 1.7 ^a^	6.0 ± 1.3 ^a^	5.5 ± 1.9 ^a^	4.9 ± 1.3 ^a^	5.4 ± 1.4 ^a^	5.8 ± 1.5 ^a^	5.1 ± 1.4 ^a^
	cluster 2	5.5 ± 1.1 ^ab^	5.5 ± 1.4 ^ab^	5.6 ± 1.6 ^ab^	4.8 ± 1.9 ^bc^	5.6 ± 1.8 ^ab^	5.7 ± 1.5 ^ab^	4.2 ± 1.7 ^c^	6.0 ± 1.4 ^a^	5.1 ± 1.5 ^abc^	5.7 ± 1.6 ^ab^
	cluster 3	5.4 ± 1.4 ^ab^	5.3 ± 1.7 ^ab^	5.2 ± 1.6 ^ab^	4.0 ± 2.0 ^c^	5.1 ± 1.6 ^abc^	5.1 ± 1.6 ^abc^	4.6 ± 1.8 ^bc^	5.9 ± 1.5 ^a^	5.6 ± 1.5 ^ab^	5.4 ± 1.5 ^ab^
	all consumers	5.4 ± 1.3 ^a^	5.4 ± 1.6 ^a^	5.4 ± 1.5 ^a^	4.5 ± 1.9 ^b^	5.4 ± 1.7 ^a^	5.4 ± 1.6 ^a^	4.5 ± 1.7 ^b^	5.9 ± 1.4 ^a^	5.4 ± 1.5 ^a^	5.5 ± 1.5 ^a^
overall flavour liking	cluster 1	6.8 ± 1.0 ^a^	6.1 ± 1.8 ^a^	5.4 ± 2.2 ^a^	7.1 ± 1.8 ^a^	6.8 ± 1.6 ^a^	5.5 ± 1.6 ^a^	5.2 ± 2.4 ^a^	7.2 ± 1.1 ^a^	7.0 ± 1.3 ^a^	6.4 ± 2.6 ^a^
	cluster 2	6.2 ± 2.0 ^abc^	5.1 ± 2.2 ^c^	6.5 ± 1.7 ^ab^	5.4 ± 2.4 ^bc^	7.0 ± 1.5 ^a^	7.3 ± 1.3 ^a^	3.0 ± 1.9 ^d^	7.1 ± 1.3 ^a^	6.5 ± 1.8 ^ab^	6.8 ± 1.6 ^a^
	cluster 3	6.7 ± 1.7 ^ab^	6.2 ± 2.1 ^abc^	5.4 ± 1.7 ^cd^	3.1 ± 1.9 ^e^	6.5 ± 2.1 ^abc^	6.4 ± 1.5 ^abc^	4.7 ± 2.5 ^d^	6.8 ± 1.5 ^a^	7.2 ± 1.4 ^a^	5.4 ± 2.2 ^bcd^
	all consumers	6.5 ± 1.8 ^ab^	5.8 ± 2.2 ^b^	5.8 ± 1.8 ^b^	4.6 ± 2.6 ^c^	6.7 ± 1.8 ^a^	6.6 ± 1.6 ^ab^	4.1 ± 2.42 ^c^	7.0 ± 1.4 ^a^	6.9 ± 1.6 ^a^	6.1 ± 2.2 ^ab^
sweet taste liking	cluster 1	6.7 ± 1.6 ^a^	7.1 ± 1.0 ^a^	6.3 ± 2.3 ^a^	7.1 ± 1.9 ^a^	6.7 ± 1.3 ^a^	5.9 ± 1.4 ^a^	6.3 ± 1.7 ^a^	7.4 ± 1.1 ^a^	7.3 ± 1.3 ^a^	6.3 ± 1.6 ^a^
	cluster 2	6.0 ± 1.8 ^ab^	5.9 ± 2.0 ^ab^	6.7 ± 1.7 ^ab^	5.6 ± 2.3 ^b^	6.6 ± 1.7 ^ab^	7.1 ± 1.2 ^a^	3.6 ± 2.2 ^c^	7.0 ± 1.1 ^a^	6.1 ± 1.8 ^ab^	6.6 ± 1.9 ^ab^
	cluster 3	7.0 ± 1.2 ^a^	6.3 ± 1.9 ^ab^	6.1 ± 1.7 ^ab^	4.2 ± 2.2 ^d^	6.4 ± 1.8 ^ab^	6.4 ± 1.7 ^ab^	4.8 ± 2.3 ^cd^	6.7 ± 1.6 ^a^	7.2 ± 1.3 ^a^	5.5 ± 1.9 ^bc^
	all consumers	6.6 ± 1.6 ^ab^	6.3 ± 1.9 ^ab^	6.4 ± 1.8 ^ab^	5.2 ± 2.4 ^cd^	6.5 ± 1.7 ^ab^	6.6 ± 1.5 ^ab^	4.6 ± 2.4 ^d^	6.9 ± 1.4 ^a^	6.8 ± 1.6 ^ab^	6.0 ± 1.9 ^bc^
margarine flavour liking	cluster 1	6.1 ± 1.8 ^a^	5.8 ± 1.8 ^a^	5.2 ± 1.5 ^a^	5.6 ± 1.6 ^a^	6.1 ± 1.2 ^a^	5.3 ± 1.5 ^a^	5.5 ± 1.8 ^a^	6.8 ± 1.5 ^a^	6.6 ± 1.4 ^a^	5.6 ± 2.3 ^a^
	cluster 2	5.6 ± 1.7 ^ab^	5.2 ± 1.7 ^b^	5.8 ± 1.5 ^ab^	5.6 ± 2.1 ^ab^	6.5 ± 1.4 ^a^	6.2 ± 1.9 ^ab^	3.6 ± 1.8 ^c^	6.4 ± 1.2 ^ab^	5.9 ± 1.4 ^ab^	6.2 ± 1.8 ^ab^
	cluster 3	6.0 ± 1.9 ^ab^	5.3 ± 1.8 ^abc^	5.0 ± 1.7 ^bcd^	4.0 ± 1.8 ^d^	5.9 ± 1.8 ^ab^	5.5 ± 2.0 ^abc^	4.5 ± 1.7 ^cd^	6.2 ± 1.8 ^ab^	6.2 ± 1.5 ^a^	5.0 ± 1.6 ^bcd^
	all consumers	5.8 ± 1.8 ^abc^	5.4 ± 1.8 ^cd^	5.4 ± 1.7 ^cd^	4.9 ± 2.0 ^de^	6.2 ± 1.6 ^ab^	5.7 ± 1.9 ^abc^	4.3 ± 1.9 ^e^	6.4 ± 1.5 ^a^	6.2 ± 1.5 ^ab^	5.6 ± 1.9 ^bcd^
hardness liking	cluster 1	5.1 ± 1.8 ^a^	5.9 ± 2.2 ^a^	4.6 ± 2.2 ^a^	5.5 ± 2.6 ^a^	5.9 ± 2.2 ^a^	4.1 ± 2.2 ^a^	5.3 ± 2.6 ^a^	6.2 ± 2.1 ^a^	5.6 ± 2.1 ^a^	5.4 ± 2.3 ^a^
	cluster 2	4.6 ± 2.3 ^b^	5.7 ± 2.1 ^ab^	5.0 ± 2.4 ^b^	4.5 ± 2.6 ^b^	5.7 ± 2.6 ^ab^	4.5 ± 2.4 ^b^	2.6 ± 1.9 ^c^	6.8 ± 1.9 ^a^	4.9 ± 2.5 ^b^	4.6 ± 2.6 ^b^
	cluster 3	5.1 ± 2.4 ^abc^	6.3 ± 1.8 ^ab^	5.1 ± 2.1 ^bc^	4.3 ± 2.5 ^c^	5.3 ± 2.3 ^abc^	4.4 ± 2.4 ^c^	3.8 ± 2.6 ^c^	6.6 ± 2.1 ^a^	6.2 ± 1.9 ^ab^	5.0 ± 2.5 ^bc^
	all consumers	4.9 ± 2.3 ^cde^	6.0 ± 2.0 ^ab^	5.0 ± 2.2 ^cde^	4.6 ± 2.6 ^def^	5.6 ± 2.4 ^bcd^	4.4 ± 2.4 ^ef^	3.6 ± 2.5 ^f^	6.6 ± 2.0 ^a^	5.6 ± 2.2 ^abc^	4.9 ± 2.5 ^cde^
crumbliness liking	cluster 1	6.2 ± 1.8 ^a^	6.3 ± 2.1 ^a^	5.9 ± 1.6 ^a^	5.4 ± 2.0 ^a^	5.8 ± 1.6 ^a^	4.8 ± 1.9 ^a^	6.5 ± 1.9 ^a^	6.4 ± 2.0 ^a^	6.3 ± 1.8 ^a^	5.8 ± 2.0 ^a^
	cluster 2	4.9 ± 2.1 ^b^	5.9 ± 1.8 ^ab^	5.8 ± 2.3 ^ab^	5.8 ± 2.7 ^ab^	6.0 ± 2.4 ^ab^	5.7 ± 2.4 ^ab^	3.8 ± 2.3 ^c^	6.2 ± 1.8 ^a^	5.6 ± 2.1 ^ab^	5.7 ± 2.3 ^ab^
	cluster 3	6.0 ± 2.0 ^ab^	6.6 ± 1.9 ^a^	5.6 ± 1.9 ^ab^	4.9 ± 2.1 ^b^	5.2 ± 2.1 ^b^	5.4 ± 1.9 ^ab^	5.2 ± 2.1 ^b^	6.6 ± 1.9 ^a^	6.6 ± 1.8 ^a^	5.4 ± 1.8 ^ab^
	all consumers	5.6 ± 2.1 ^abc^	6.3 ± 1.9 ^ab^	5.7 ± 2.0 ^abc^	5.4 ± 2.3 ^bc^	5.6 ± 2.2 ^abc^	5.4 ± 2.1 ^bc^	4.9 ± 2.3 ^c^	6.4 ± 1.9 ^a^	6.2 ± 2.0 ^ab^	5.6 ± 2.0 ^abc^
overall liking	cluster 1	6.6 ± 1.4 ^a^	6.3 ± 1.7 ^a^	5.4 ± 2.4 ^ab^	7.1 ± 1.9 ^a^	7.3 ± 1.1 ^a^	3.9 ± 2.0 ^b^	5.9 ± 2.3 ^ab^	6.4 ± 1.5 ^a^	6.6 ± 1.4 ^a^	5.7 ± 2.0 ^ab^
	cluster 2	5.8 ± 2.0 ^bc^	5.3 ± 2.2 ^c^	6.7 ± 1.6 ^ab^	5.7 ± 2.4 ^bc^	6.6 ± 1.7 ^abc^	6.4 ± 1.7 ^abc^	2.5 ± 1.4 ^d^	7.2 ± 1.4 ^a^	5.9 ± 2.0 ^abc^	6.8 ± 1.7 ^ab^
	cluster 3	6.6 ± 1.7 ^ab^	6.4 ± 2.3 ^ab^	5.6 ± 1.8 ^bc^	3.0 ± 1.7 ^e^	6.2 ± 1.9 ^abc^	5.9 ± 1.8 ^abc^	4.3 ± 2.3 ^d^	6.9 ± 1.8 ^a^	7.1 ± 1.7 ^a^	5.0 ± 2.1 ^cd^
	all consumers	6.3 ± 1.8 ^ab^	5.9 ± 2.2 ^b^	6.0 ± 1.9 ^b^	4.7 ± 2.6 ^c^	6.5 ± 1.7 ^ab^	5.8 ± 2.0 ^b^	3.8 ± 2.3 ^c^	6.9 ± 1.6 ^a^	6.6 ± 1.8 ^ab^	5.8 ± 2.1 ^b^

Data are presented as mean ± standard deviation (*n* = 100). Different letters represent significant difference between means within each row (*p* < 0.05; ANOVA test, Tukey-HSD post hoc test).

**Table 11 foods-12-00018-t011:** Consumers’ purchase intention for biscuits.

Biscuit	Not Buying (WDNB + WPNB)	Not Sure	Buying (WPB + WDB)
control	31 (12 + 19) ^a^	30 ^b^	39 (31 + 8) ^bc^
rapeseed 2%	30 (12 + 18) ^a^	26 ^ab^	44 (26 + 18) ^bcd^
rapeseed 5%	26 (9 + 17) ^a^	30 ^b^	44 (33 + 11) ^bcd^
rapeseed 10%	62 (42 + 20) ^bc^	15 ^ab^	23 (12 + 11) ^ab^
phacelia 2%	27 (8 + 19) ^a^	14 ^ab^	59 (35 + 24) ^cd^
phacelia 5%	39 (15 + 24) ^ab^	28 ^ab^	33 (24 + 9) ^bc^
phacelia 10%	83 (63 + 20) ^c^	7 ^a^	10 (3 + 7) ^a^
sunflower 2%	16 (7 + 9) ^a^	16 ^ab^	68 (36 + 32) ^d^
sunflower 5%	22 (5 + 17) ^a^	31 ^b^	47 (33 + 14) ^bcd^
sunflower 10%	35 (18 + 17) ^ab^	25 ^ab^	40 (26 + 14) ^bc^

WDNB—would definitely not buy; WPNB—would probably not buy; WPB—would probably buy; WDB—would definitely buy. Values in the same coloumn marked with different letter in superscript correspond to significant difference between samples according the Chi-square test (*p* < 0.05) followed by Marascuilo procedure.

## Data Availability

The data that support the findings of this study are available from the corresponding author upon reasonable request.

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
