# Peer review of "Biscuits Enriched with Monofloral Bee Pollens: Nutritional Properties, Techno-Functional Parameters, Sensory Profile, and Consumer Preference"

_foods, 2022, doi:10.3390/foods12010018_

Round 1

Reviewer 1 Report

Very nice work, congrats to authors.

Here are my suggestions:

General comment: Please avoid to use "our". There are no "I", "my", "we" and "our" in scientific language. It should be avoided as much as possible.

Line 40: Delete "The" at the beginning of sentence. It is surplus.

Line 44: It should be "Apis mellifera L.". Please correct.

Line 54: Please replace term "minerals" to be "macro and microelements". Also, it should be "vitamins B complex" in plural. There are several vitamins in this group.

Line 80: Delete "and". It is surplus here.

Lines 90-94: Please specify on what basis you identify pollen grains? What was your referent models?

Line 126: Please replace term "polyphenols" with "phenolic". Polyphenols are flavonoids and other big structures while phenolics gathered also phenolic acids as simple phenolic compounds. Please correct and replace this term through a whole text.

Lines 143-144: Please specify "mg" for AAE as well as micromols or mg for Trolox here?

Line 309: It should be "Total phenolic content and antioxidant properties"

Line 320: It can not be "thousand" mg, it is impossible. It can be thousand micrograms maybe? Please check/correct.

Lines 324-328: Authors missed here very important factor that can influence on bioavailability or bioaccessibility of phenolic compounds and that are interactions with other macronutrients presented in matrix like proteins, lipids and sugars. Please see review (https://doi.org/10.1016/j.foodchem.2014.12.013) as well as article about interactions in food product enriched with bee pollen (https://doi.org/10.1016/j.foodchem.2021.129310 ).

Line 393: typo - correct unit cm3 here.

Lines 606-608: As we saw through Manuscript acceptance of cookies by consumers was also strongly influenced with botanical origin of pollen. It should be also mentioned in Conclusion section.

Kind regards.

Author Response

Answers for review of Manuscript ID: foods-2103936 submitted to Foods entitled

” Biscuits enriched with monofloral bee pollens: nutritional properties, techno-functional parameters, sensory profile, and consumer preference”

First, we would like to thank your work of reading through the manuscript and giving us suggestions to improve the quality of the publication. Our research team has carefully read through your recommendations and upon our best knowledge we’ve made the corrections. We have also checked the English spelling and made corrections. Our answers are listed below.

Reviewer #1

General comment: Please avoid to use "our". There are no "I", "my", "we" and "our" in scientific language. It should be avoided as much as possible.

Response: Usage of the word ”our” has been minimized in the text.

Line 40: Delete "The" at the beginning of sentence. It is surplus.

Response: The superfluous word has been removed.

Line 44: It should be "Apis mellifera L.". Please correct.

Response: The designation has been corrected.

Line 54: Please replace term "minerals" to be "macro and microelements". Also, it should be "vitamins B complex" in plural. There are several vitamins in this group.

Response: The terms have been corrected.

Line 80: Delete "and". It is surplus here.

Response: A missing word has been added the text.

Lines 90-94: Please specify on what basis you identify pollen grains? What was your referent models?

Response: A brief description was added on the identificaton method of pollen grains.

Line 126: Please replace term "polyphenols" with "phenolic". Polyphenols are flavonoids and other big structures while phenolics gathered also phenolic acids as simple phenolic compounds. Please correct and replace this term through a whole text.

Response: The term has been corrected through the whole manuscript.

Lines 143-144: Please specify "mg" for AAE as well as micromols or mg for Trolox here?

Response: The unit of measurement (mg) was added to the text.

Line 309: It should be "Total phenolic content and antioxidant properties"

Response: The title has been modified.

Line 320: It can not be "thousand" mg, it is impossible. It can be thousand micrograms maybe? Please check/correct.

Response: The information has been clarified.

Lines 324-328: Authors missed here very important factor that can influence on bioavailability or bioaccessibility of phenolic compounds and that are interactions with other macronutrients presented in matrix like proteins, lipids and sugars. Please see review (https://doi.org/10.1016/j.foodchem.2014.12.013) as well as article about interactions in food product enriched with bee pollen (https://doi.org/10.1016/j.foodchem.2021.129310 ).

Response: Thank you very much for the suggested articles which included relevant information. The necessary information has been added to the text.

Line 393: typo - correct unit cm3 here.

Response: The typo has been corrected.

Lines 606-608: As we saw through Manuscript acceptance of cookies by consumers was also strongly influenced with botanical origin of pollen. It should be also mentioned in Conclusion section.

Response: The conclusion has been completed with the necessary information.

Reviewer 2 Report

The manuscript is interesting but should be improved. I have some remarks mentioned below

- Please rewrite and organize the abstract according to the following context:

A short introduction, hypothesis (aim) of the study, methods, the most important quantitative results (provide numbers), a general conclusion, and future prospective

  - The authors should include some information about biscuits and why they have selected it in the introduction section. Please provide this info to the readers.

- Page 2, Lines 85-94, please provide a reference

- Page 3, Line 99. What informed the use of 2, 5, and 10% bee pollens powder in biscuits production?

-Page 3, Line 124, (eq1) seems good but where is the fiber content?

-Page 4, Line 132, please provide the city and country of manufacture.

-Page 4, Lines 134 -144, this section is confusing and very short, so please reformat it with more detail.

-Statistical analysis, this section is very long, please reduce it.

- Please add (p < 0.05) when “significant differences” are mentioned anywhere in the text.

- The effect of varying concentrations of pollen is not discussed well, why?

- Figure 2, should move to the supplementary materials file.

 - Conclusions section, please highlight the future standpoint well.

- Please care about scientific names, should be in italic throughout the MS.

-  Manuscript has grammatical errors, please check.

Author Response

Answers for review of Manuscript ID: foods-2103936 submitted to Foods entitled

” Biscuits enriched with monofloral bee pollens: nutritional properties, techno-functional parameters, sensory profile, and consumer preference”

First, we would like to thank your work of reading through the manuscript and giving us suggestions to improve the quality of the publication. Our research team has carefully read through your recommendations and upon our best knowledge we’ve made the corrections. We have also checked the English spelling and made corrections. Our answers are listed below.

Reviewer #2

- Please rewrite and organize the abstract according to the following context: A short introduction, hypothesis (aim) of the study, methods, the most important quantitative results (provide numbers), a general conclusion, and future prospective

Response: The abstract was modified according to the above suggestions.

- The authors should include some information about biscuits and why they have selected it in the introduction section. Please provide this info to the readers.

Response: A brief description about biscuits was added, and and an explanation was provided.

- Page 2, Lines 85-94, please provide a reference

Response: The relevant reference was added.

- Page 3, Line 99. What informed the use of 2, 5, and 10% bee pollens powder in biscuits production?

Response: Explanation was provided in the text. 2, 5, and 10% of substitution were applied to obtain results that are comparable with the data in the literature.

-Page 3, Line 124, (eq1) seems good but where is the fiber content?

Response: The fibre content of biscuits was not determined. The carbohydrate content includes both digestible and non-digestible carbohydrates.

-Page 4, Line 132, please provide the city and country of manufacture.

Response: Details were provided about the manufacturer of the centrifuge.

-Page 4, Lines 134 -144, this section is confusing and very short, so please reformat it with more detail.

Response: Detailed descriptions of the spectrophotometric methods have been added to the text.

-Statistical analysis, this section is very long, please reduce it.

Response: Section 2.10. (Statistical analysis) has been reduced.

- Please add (p < 0.05) when “significant differences” are mentioned anywhere in the text.

Response: As the level of significance was defined in the ”Statistical analysis” section and was also indicated in the footnotes of tables, we consider it unnecessary to add (p < 0.05) it in each sentence when ”Significant difference” is mentioned.

- The effect of varying concentrations of pollen is not discussed well, why?

Response:

Our research has focused on to detail all the measurement results for each parameter in tables, where of course the statistical calculations are also shown. In addition, figures have been prepared to make the effect of pollen enrichment even more illustrative. As suggested, we have complemented the manuscript: antioxidant capacity, colour

- Figure 2, should move to the supplementary materials file.

Response: A supplementary matierial file containing the figure of penalty analysis was created.

- Conclusions section, please highlight the future standpoint well.

Response: The future standpoint has been clarified.

- Please care about scientific names, should be in italic throughout the MS.

Response: Each of the scientific names has been formatted to italic. 

-  Manuscript has grammatical errors, please check.

Response: Grammatical errors have been minimized in the text.

Reviewer 3 Report

In the current article, the effects of enrichment with monofloral bee pollens on the nutritional characteristics, techno-functional factors, and sensory properties of biscuits were studied. In my opinion, the paper has a good structure and has innovation and appropriate examinations. The article could be minor revisions as follow:

L34: Write the suggestion (recommendation) of the findings of this research in one sentence at the end of the abstract.

L35:  remove some keywords.

L40: remove "The".

L47: Do you know the economic value and price of pollen?

L54: add comma (,) before "and vitamin E".

L77: add comma (,) before "and baking soda".

L79: Bee pollens were obtained from which region and in which season?

L103: Did the biscuits cooling room temperature? Was a fan used?

L113: the caption of Figure 1 is incomplete.

L131: write the temperature and power of ultrasonic bath.

L197: Accessors?????

L204: change "assessors" to "panelists". Check and edit in the text.

L208: Do the same as the previous comment.

L294: change "water" to "moisture content".

Table 2: change "water" to "moisture content".

Table 2: add (%) in the title of the table.

3.2.2. Write the effect of increasing the replacement rate of pollen with flour on the antioxidant capacity of the samples.

3.4. the baking loss (%) of samples included 2 and 10% phacelia were significantly higher than other samples. Why?

Author Response

Answers for review of Manuscript ID: foods-2103936 submitted to Foods entitled

” Biscuits enriched with monofloral bee pollens: nutritional properties, techno-functional parameters, sensory profile, and consumer preference”

First, we would like to thank your work of reading through the manuscript and giving us suggestions to improve the quality of the publication. Our research team has carefully read through your recommendations and upon our best knowledge we’ve made the corrections. We have also checked the English spelling and made corrections. Our answers are listed below.

Reviewer 3

L34: Write the suggestion (recommendation) of the findings of this research in one sentence at the end of the abstract.

Response: A suggestion was added to the abstract.

L35:  remove some keywords.

Response: Two of the keywords have been removed.

L40: remove "The".

Response: The superfluous word has been removed.

L47: Do you know the economic value and price of pollen?

Response: Information on the economic aspects of bee pollen is very limited, but it has been added to the text that the bee pollen market is growing.

L54: add comma (,) before "and vitamin E".

Response: Comma has been added.

L77: add comma (,) before "and baking soda".

Response: Comma has been added.

L79: Bee pollens were obtained from which region and in which season?

Response: Information on the origin of pollen pellets has been provided.

L103: Did the biscuits cooling room temperature? Was a fan used?

Response: Biscuits were cooled at room temperature. This infomation has been added to the methods.

113: the caption of Figure 1 is incomplete.

Response: The caption of Figure 1 has been completed with ”after baking”

L131: write the temperature and power of ultrasonic bath.

Response: Description of extraction conditions was completed with the above-mentioned parameters.

L197: Accessors?????

Response: The typo has been corrected.

L204: change "assessors" to "panelists". Check and edit in the text.

Response: The usage of the term ”assessors” has been minimized.

L208: Do the same as the previous comment.

Response: The usage of the term ”assessors” has been minimized.

L294: change "water" to "moisture content".

Response: The term has been corrected.

Table 2: change "water" to "moisture content".

Response: The term has been corrected.

Table 2: add (%) in the title of the table.

Response: The percentage mark has been indicated.

3.2.2. Write the effect of increasing the replacement rate of pollen with flour on the antioxidant capacity of the samples.

Response: Effects of pollen substitution on the antioxidant capacity of samples were detailed in section 3.2.2.

3.4. the baking loss (%) of samples included 2 and 10% phacelia were significantly higher than other samples. Why?

Response: All biscuit samples were made under same conditions (same oven, same temperature, same baking time). The only difference between the biscuit samples is the difference between the replicates, this could be due to several factors: probably the phacelia pollen dosage increases the weight loss, but at the same time the sensitivity of the statistical post-hoc tests is different (Dunn, Conover-Iman, Steel-Dwass-Critchlow-Fligner, etc.), we used Dunn's pairwise procedure with Bonferroni correction in our tests here.

Round 2

Reviewer 2 Report

The MS has been significantly improved.